# Robustness in Large Language Models: A Survey of Mitigation Strategies and Evaluation Metrics

**Pankaj Kumar**                                    *pankaj.kumar@niser.ac.in*
*School of Computer Sciences*
*National Institute of Science Education and Research*
*An OCC of Homi Bhabha National Institute, India*

**Subhankar Mishra**                                 *smishra@niser.ac.in*
*School of Computer Sciences*
*National Institute of Science Education and Research*
*An OCC of Homi Bhabha National Institute, India*

**Reviewed on OpenReview:** *https://openreview.net/forum?id=Bchvaaod6g*

## Abstract

Large Language Models (LLMs) have emerged as a promising cornerstone for the development of natural language processing (NLP) and artificial intelligence (AI). However, ensuring the robustness of LLMs remains a critical challenge. To address these challenges and advance the field, this survey provides a comprehensive overview of current studies in this area. First, we systematically examine the nature of robustness in LLMs, including its conceptual foundations, the importance of consistent performance across diverse inputs, and the implications of failure modes in real-world applications. Next, we analyze the sources of non-robustness, categorizing intrinsic model limitations, data-driven vulnerabilities, and external adversarial factors that compromise reliability. Following this, we review state-of-the-art mitigation strategies, and then we discuss widely adopted benchmarks, emerging metrics, and persistent gaps in assessing real-world reliability. Finally, we synthesize findings from existing surveys and interdisciplinary studies to highlight trends, unresolved issues, and pathways for future research.

## 1 Introduction

Large Language Models (LLMs), mainly characterized by their vast number of parameters, have emerged as a promising cornerstone for the development of natural language processing(NLP) and artificial intelligence (AI) Zhang et al. (2023). These models have demonstrated remarkable capabilities across a wide range of NLP applications. Foundation models like GPT-4 OpenAI et al. (2024), GPT-3 Brown et al. (2020), Qwen 2.5-VL Bai et al. (2025a), Deepseek-V3 DeepSeek-AI et al. (2025), Meta (LLaMA-3, LLaMA-2 Touvron et al. (2023b), LLaMA-1 Touvron et al. (2023a)), Gemini DeepMind (2023) are pre-trained on massive datasets and serve as the backbone for various AI tasks, from text generation to multimodal understanding—the ability to process and interpret multiple types of data, such as text and images. Their extensive factual knowledge has significantly improved their effectiveness in information retrieval, enabling them to provide more accurate and contextually relevant responses AlKhamissi et al. (2022); Petroni et al. (2019). As a result, these models achieve high accuracy on benchmark evaluations, demonstrating their strong predictive capabilities across diverse tasks OpenAI et al. (2024); Bai et al. (2025a); DeepSeek-AI et al. (2025); Grattafiori et al. (2024). However, accuracy alone is insufficient to ensure reliability in real-world applications.

In recent years, despite improvements in the accuracy of deep neural network models, researchers have found that they are easy to fool by applying a specific imperceptible perturbation Szegedy et al. (2014). A deep learning model's accuracy describes how accurately the model predicts sample points over a distribution. If

a model cannot predict accurately, then no matter what properties this model holds, it would be meaningless. Although accuracy is the most fundamental metric in the evaluation and is essential to validate the performance of an LLM before it is released, it is equally important to test its robustness when deploying for the development of reliable AI systems. Robustness, a crucial aspect of deep learning models, describes their ability to maintain stable predictions when faced with specific perturbations or variations in the input data Liu et al. (2024b). Despite recent advancements, ensuring the robustness of LLMs remains a critical challenge. LLMs may encounter unpredictable variations in language, shifts in data (changes in the data patterns that a model encounters, which may affect its performance), and adversarial inputs that can affect their performance. In practical applications, these models often face noisy, unstructured text, breaking the assumption that input data will always be clean and well-formed. Additionally, a lack of robustness may result in unintended biases, incorrect predictions, or overreliance on spurious correlations (misleading correlations in data that a model may rely on incorrectly), raising concerns about their reliability in high-stakes environments such as healthcare Wan et al. (2024b;a); He et al. (2025), law Lee (2023); Cui et al. (2024); Yue et al. (2023); Bhambhoria et al. (2024); Bai et al. (2021b), complex reasoning Yang et al. (2023), code generation Sun et al. (2024b); Lin et al. (2024a); Mishra et al. (2024); Hassid et al. (2024), and finance Lee et al. (2025); Zhao et al. (2024a); Li et al. (2024c); Wu et al. (2023). Because of this, a thorough and exacting assessment of LLM robustness that goes beyond conventional accuracy criteria is required.

However, advancing research on the robustness of LLMs also faces significant challenges. First, the absence of standardized definitions and categorizations of robustness complicates cross-study comparisons, as research efforts often differ in how they frame their scope and components. Second, pinpointing the root causes of non-robustness, such as data biases, unstable training dynamics, architectural limitations, or inference inefficiencies, requires disentangling complex interactions across these factors. Third, while numerous mitigation strategies exist, their effectiveness often depends on addressing specific vulnerabilities, making it difficult to generalize solutions. Fourth, evaluating robustness remains fragmented, with no unified metrics or benchmarks to holistically assess performance under diverse real-world conditions like noisy inputs or adversarial attacks. Finally, existing literature on LLM robustness (Zhao et al., 2025; Gu et al., 2025; He et al., 2025; Liu et al., 2024b; Mehrabi et al., 2022; Wang et al., 2025; Lu et al., 2024; Weng, 2023; Yang et al., 2024a) is scattered across domains, limiting cross-pollination of insights and hindering the development of cohesive solutions.

To address these challenges and advance the field, this survey provides a comprehensive overview of current studies in this area. Figure 1 shows the outline of this survey. We begin by clarifying definitions and categorizing the dimensions of robustness to establish a shared framework for analysis (§2). Next, we systematically identify and classify the primary sources of non-robustness, including data quality issues, training instabilities, architectural constraints, and inference-time vulnerabilities (§3). We then review state-of-the-art mitigation strategies, mapping each approach to the specific weaknesses it targets (§4). Following this, we analyze evaluation methodologies, discussing widely adopted benchmarks, emerging metrics, and persistent gaps in assessing real-world reliability (§5). Finally, we synthesize findings from existing surveys and interdisciplinary studies to highlight trends, unresolved issues, and pathways for future research (§6). By unifying these perspectives, we aim to foster collaboration and accelerate progress toward building more reliable, resilient LLMs.

The existence of numerous surveys, while indicating intense research activity, also suggests a potentially fragmented landscape where different communities might use varying terminology or focus on specific aspects of the broader robustness problem. A key contribution of a comprehensive survey like this one is to bridge these perspectives and synthesize the common underlying principles and challenges. Furthermore, existing surveys often highlight gaps and open questions that remain pertinent. The lack of unified definitions and practical frameworks for operationalising trustworthiness (including robustness) is noted (de Cerqueira et al., 2025). The difficulty in evaluating generative tasks robustly is frequently mentioned. Understanding and mitigating more subtle or complex shortcuts beyond simple lexical cues remains an area for research (Howe et al., 2025; Zhou et al., 2024d; Yuan et al., 2023). Finally, ensuring that robustness improvements scale effectively with increasing model size is an ongoing concern.

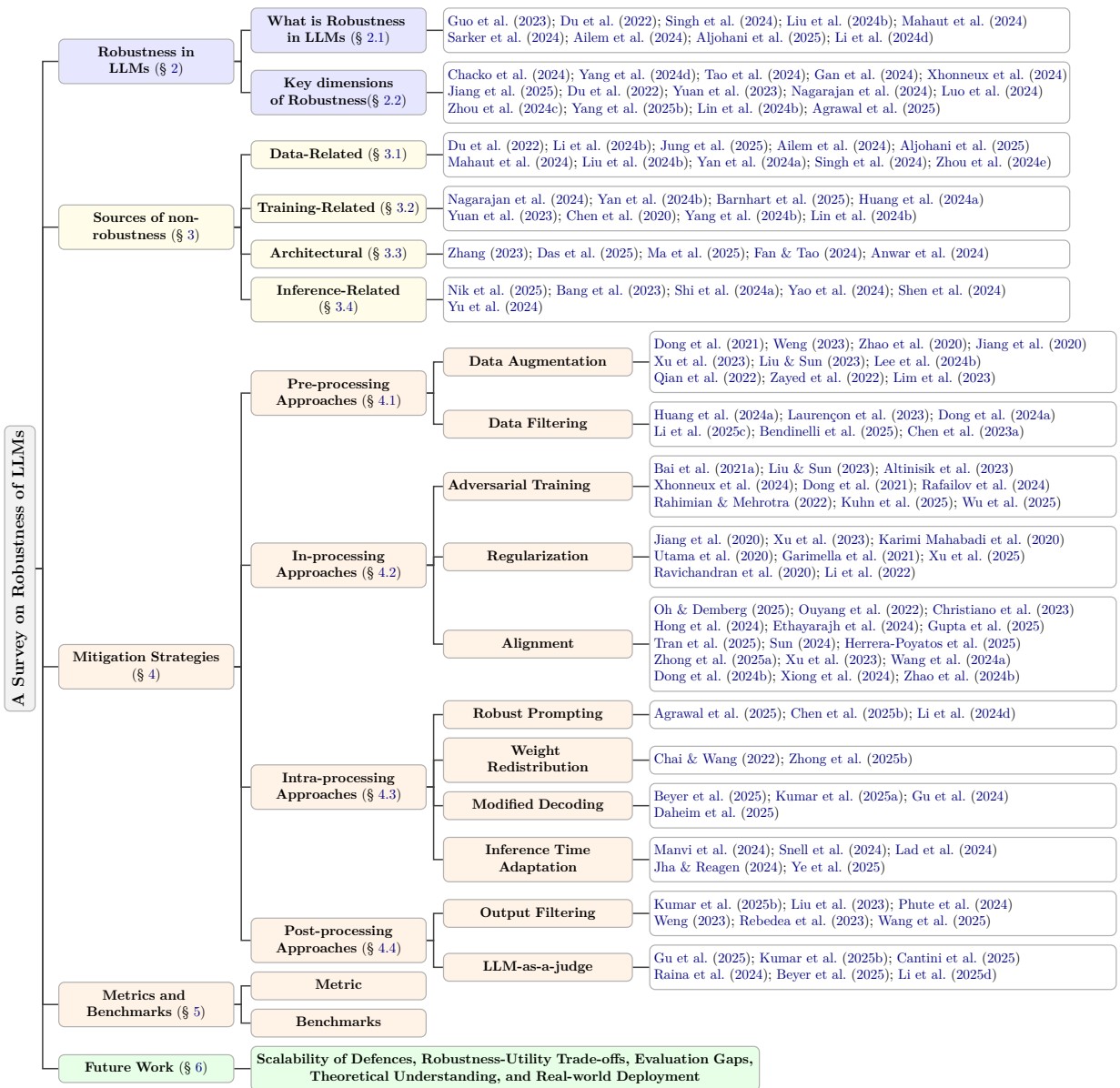

Figure 1: The outline of the survey on robustness of LLMs.

## 1.1 Search Methodology

The purpose of this survey is to gather and categorize research work that helps to improve robustness in LLMs. A systematic search was conducted across multiple digital libraries and repositories to identify relevant studies on robustness in LLMs as shown in Fig. 2. Table 1.1 summarizes the databases and sources that were included:

This search covered studies published between **January 2020 and June 2025**, ensuring inclusion of the most recent work on LLM robustness. Search strings were tailored to each database but generally combined terms related to "large language models", "robustness", "evaluation", and "mitigation strategies". A representative query structure was:

```
("large language model" OR "LLM" OR "foundation model")
AND (robustness OR reliable OR adversarial OR OOD OR evaluation OR mitigating robustness)
```

Table 1: Search results by source

| Source | Identified | Screened | Included |
|---|---|---|---|
| arXiv | 389 | 258 | 173 |
| ACL Anthology | 47 | 35 | 21 |
| ACM Digital Library | 26 | 15 | 5 |
| IEEE Xplore | 17 | 7 | 1 |
| OpenReview | 22 | 12 | 7 |
| GitHub | 7 | 4 | 2 |
| **Total** | **508** | **337** | **209** |

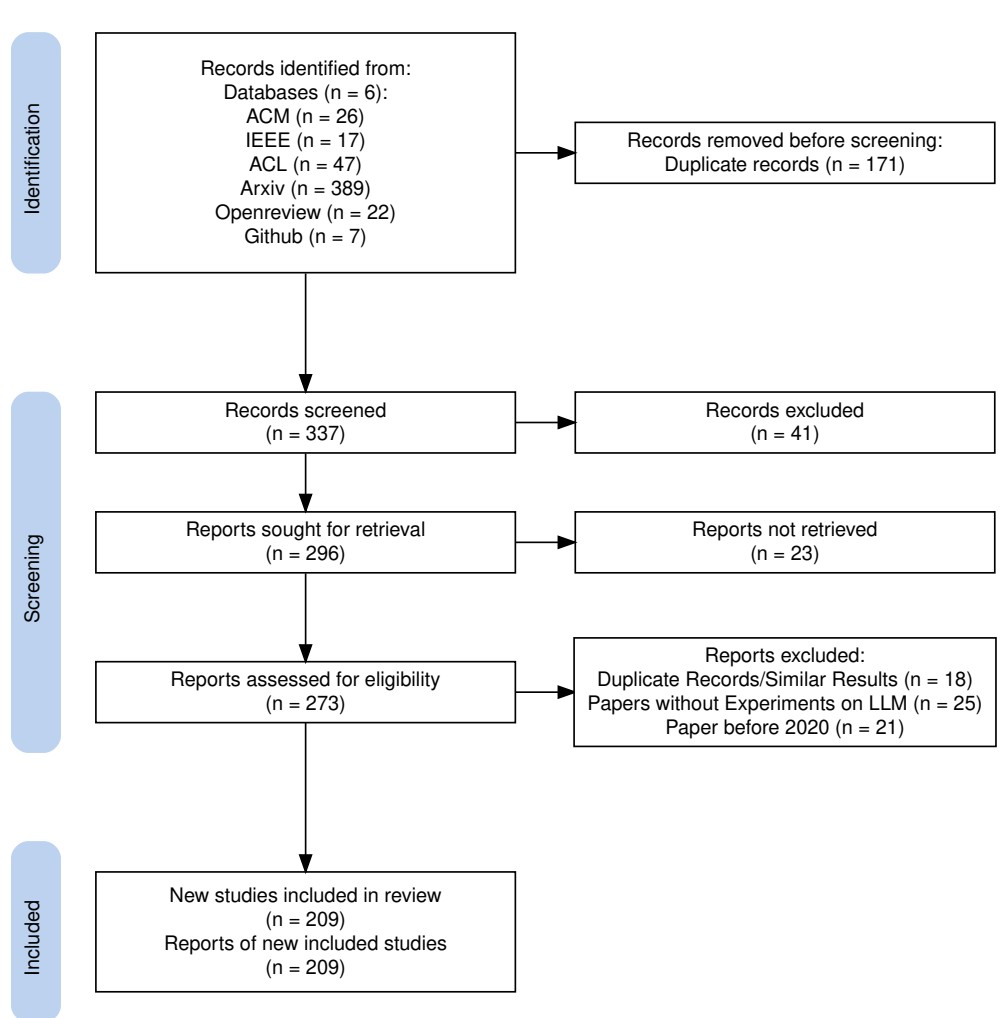

Figure 2: PRISMA flow diagram of the survey methodology

### 1.1.1 Inclusion Criteria

Studies were included if they met the following criteria:

- Explicitly addressed **robustness, reliability, or evaluation** of LLMs.

- Contained **empirical experiments** or benchmarks involving LLMs.

- Published within the time range **2020–2025**.

### 1.1.2 Exclusion Criteria

Studies were excluded if they:

- Were duplicates or near-duplicate entries across sources or with similar concepts or results.

- Did not contain experiments on LLMs (e.g., purely theoretical works, works on VLM).

- Were published before 2020.

- Were inaccessible or unretrievable.

Further, after completing this criterion, we applied backward snowballing (i.e., finding new publications by tracing references cited in this paper with some relevance). This process was repeated iteratively for each newly identified publication.

## 2 Robustness in LLMs

In the context of LLMs from various literatures, *robustness* refers to their ability to perform reliably and safely despite challenges like input variations, unexpected data, or adversarial attacks (which involve crafting specific inputs designed to mislead the model into producing incorrect or unintended outputs, despite the inputs appearing normal and being susceptible to humans). It is frequently discussed as a critical component within the broader framework of AI Trustworthiness. Trustworthy AI systems are expected to be reliable, safe, fair, transparent, accountable, and robust. Robustness underpins several of these qualities; for instance, a model prone to adversarial manipulation (lack of robustness) cannot be considered safe or reliable (Wang et al., 2023a; Aljohani et al., 2025). Similarly, a model that hallucinates or provides inconsistent answers under slightly different prompts lacks reliability, another facet closely tied to robustness (Zhou et al., 2024c). Research into the vulnerabilities of machine learning models, including LLMs, has often focused on adversarial robustness - the model's resilience against inputs intentionally crafted to cause misclassification or undesired behaviour (Zhao et al., 2025).

This survey approaches robustness comprehensively, considering LLMs as integrated systems rather than isolated components.

### 2.1 What is Robustness in LLMs

The concept of robustness for LLMs is multifaceted, and various studies approach its definition from different angles, reflecting the diverse ways models can fail or exhibit instability. Formally, robustness can be conceptualized as the degree to which a model's output $f(x)$ remains stable or correct when the input $x$ is subjected to some form of perturbation $\delta$ or drawn from a distribution $\mathcal{D}_{test}$ different from the training distribution $\mathcal{D}_{train}$. The specific nature of the perturbation or distribution shift defines the dimension of robustness being considered. This concept is vital for setting research goals and evaluation standards, as robustness directly impacts whether users can trust these models in real-world applications. Synthesizing across the literature, several core themes emerge:

- Stability under Disruption: Robustness is fundamentally about the stability of LLM performance and behaviour when faced with various forms of disruption, including unseen scenarios, different types of attacks, and noise in the input (Guo et al., 2023). It implies predictability even when conditions deviate from the ideal.

- Consistency and Reliability of Outputs: Robustness requires the model to consistently generate outputs that are accurate, reliable, and unbiased across diverse scenarios (Lad et al., 2024; Aljohani

et al., 2025). This involves minimizing errors, factual inaccuracies (hallucinations), and harmful biases, even under challenging conditions.

- Adherence to Intended Behaviour: A robust model should adhere to its intended function and safety constraints. This includes following legitimate instructions correctly while resisting malicious manipulations, such as prompt injection attacks (Li et al., 2024d), (which involves performing unintended actions by combining adversarially designed malicious input with the model's instructions) or backdoor attacks (Cai et al., 2022; Chen et al., 2021).

- Resilience to Input Variations: A robust LLM should maintain consistent outputs despite variations in the input that preserve the core meaning or intent. This includes resilience to natural language variations like morphological changes, typos, paraphrasing, and syntactic restructuring (Singh et al., 2024; Mahaut et al., 2024; Sarker et al., 2024). It also encompasses stability when confronted with unforeseen or malformed prompts (Ailem et al., 2024).

- Performance Maintenance under Distribution Shifts (Du et al., 2022): Robustness entails maintaining performance levels when the distribution of deployment data differs from the training data (OOD generalization). This is crucial as real-world data rarely perfectly matches training sets.

Drawing these threads together, a working definition for this survey is:

> **LLM's robustness** refers to the model's ability to maintain consistent performance, reliability, and adherence to intended behaviour (including accuracy, factuality, safety, and fairness) despite variations in inputs, contexts, or underlying data distributions. This encompasses resilience to natural noise, semantic paraphrasing, distribution shifts, incomplete information, reasoning perturbations, instruction variations, and adversarial manipulations.

## 2.2 Key Dimensions of Robustness

LLM robustness is not a monolithic property but rather a collection of related capabilities. To better understand and address the challenge, it is useful to delineate the key facets of LLM robustness, recognizing that these categories often overlap and interact. Figure 3 provides an overview of this overlapping nature, illustrating how progress in one dimension can amplify or undermine robustness in others due to their inherent interdependencies.

### 2.2.1 Resilience to Adversarial Attacks

It refers to a model's ability to withstand inputs that have been intentionally manipulated by an adversary to trigger specific failures (Chacko et al., 2024). Such perturbations are typically small, often imperceptible or semantically plausible to humans, yet they exploit latent vulnerabilities in the model. These attacks commonly manifest as misclassifications in classification tasks (Yang et al., 2024d), the generation of harmful or toxic content, or the circumvention of safety alignments through jailbreaks (a class of prompt injections designed to override safety filters and moderation policies) (Tao et al., 2024). In broader cases, adversarial perturbations can simply degrade model performance without obvious failure.

Attacks on LLMs vary in their granularity. At the input level, adversaries can manipulate characters (e.g., HotFlip), words (e.g., synonym substitution), or entire sentences (e.g., insertion of distracting phrases) (Yang et al., 2024d). More subtle perturbations such as typographical errors have also been shown to degrade reasoning robustness (Gan et al., 2024). Attacks further differ depending on the threat model: white-box attacks assume access to internal parameters or gradients, while black-box attacks rely solely on interacting with inputs and outputs. Prominent techniques include Greedy Coordinate Gradient (GCG), AutoDAN, and PAIR (Xhonneux et al., 2024), with some methods targeting continuous embeddings rather than discrete tokens. In the case of multimodal LLMs (MLLMs), adversaries can exploit vulnerabilities within individual modalities (e.g., image or text) or across modality interactions, compounding the risk (Jiang et al., 2025).

At their core, adversarial attacks succeed by exploiting brittle statistical dependencies and spurious correlations learned during training, exposing the fragility of LLMs to small but strategically chosen perturbations.

### 2.2.2 OOD Genralization

It concerns the ability of a model to sustain performance when faced with inputs drawn from distributions different from those encountered during training. Real-world deployment frequently involves such distribution shifts, arising from temporal drift, novel domains, or different user populations. When tested on OOD datasets, LLMs often experience significant performance degradation relative to in-distribution (ID) benchmarks (Du et al., 2022). Failures commonly emerge from covariate shift, where input features differ, or concept shift, where input–output mappings change (Yuan et al., 2023; Nagarajan et al., 2024).

These challenges surface in multiple forms: difficulties with unfamiliar knowledge domains, stylistic shifts in text, or multilingual and code-mixed settings. Such failures reflect a tendency to rely on spurious correlations and biases present in training data, rather than learning generalizable, invariant features.

### 2.2.3 Sensitivity to Prompt Variations

It captures how much a model's output changes when the input instruction is rephrased or slightly altered while retaining its semantic intent (Yang et al., 2024d). Unlike humans, who are generally resilient to paraphrasing or minor structural shifts, LLMs can exhibit drastic changes in correctness, coherence, or consistency under such conditions (Agrawal et al., 2025).

Even simple alterations such as synonym replacement, insertion or deletion of words, or reordering of answer options can destabilize performance. This fragility often stems from overfitting to prompt formats encountered during training and reliance on superficial cues rather than deeper semantic understanding. Reducing sensitivity to prompt variation is therefore essential for achieving robustness in instruction-following scenarios.

### 2.2.4 Handling Noisy or Corrected Input

It refers to a model's capacity to process data containing imperfections commonly found in real-world use. Such noise may include typographical errors, misspellings, irregular grammar, or extraneous punctuation. Automated pipelines introduce additional errors, e.g., transcription mistakes from voice-to-text systems or inaccuracies from OCR (Optical Character Recognition) processes.

Models trained or fine-tuned on noisy data are further at risk, as they may learn brittle mappings that amplify fragility rather than tolerance (Luo et al., 2024). In practice, noisy or corrupted inputs can distract the model and degrade downstream task performance, underscoring the need for architectures and training regimes that explicitly account for the messiness of real-world data.

### 2.2.5 Fairness Under Stress

It highlights a model's ability to uphold fairness and avoid amplifying societal biases when confronted with adversarial prompts or challenging contexts (Jung et al., 2025). Failures in this dimension manifest as disparities in prediction quality across socio-demographic or geographic groups. For instance, LLMs often perform better on U.S.-based datasets than on Chilean ones, reflecting geographical bias. In the U.S., attributes such as race and political identity influence accuracy, whereas in Chile, gender, education, and religious affiliation play stronger roles (Abeliuk et al., 2025; Qu & Wang, 2024).

These disparities typically originate from imbalances in training data, where certain populations are underrepresented or misrepresented, resulting in skewed learned distributions. Addressing this requires approaches that ensure equitable performance across diverse populations, even under adversarial or distributionally shifted conditions.

### 2.2.6 Consistency and Reliability of Outputs

Consistency and reliability pertain to the stability and trustworthiness of a model's generations across semantically similar inputs. Failures of consistency include producing contradictory answers to paraphrased queries or generating factually incorrect, nonsensical, or hallucinated content (Zhou et al., 2024c; Yang et al., 2025b).

These issues arise from a lack of stable internal representations and an overemphasis on local statistical fluency at the expense of factual soundness or logical coherence. As such, inconsistency is closely tied to other robustness dimensions, including OOD generalization and adversarial susceptibility. Ensuring consistency is not only a matter of reliability but also a prerequisite for deploying LLMs in domains where factual accuracy is critical.

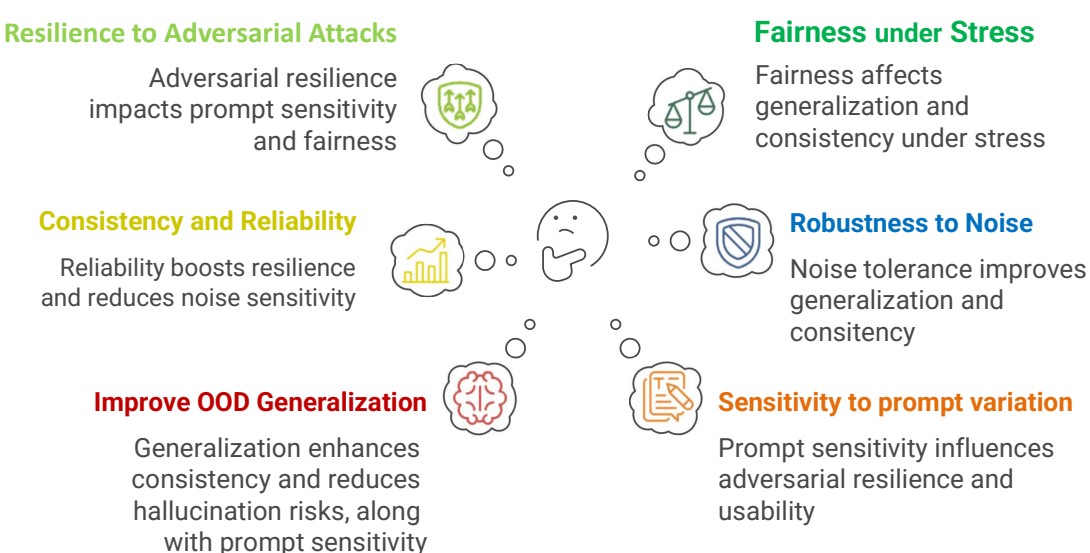

**Enhance LLM Robustness: Synergies and Tradeoffs**

**Resilience to Adversarial Attacks**
Adversarial resilience impacts prompt sensitivity and fairness

**Fairness under Stress**
Fairness affects generalization and consistency under stress

**Consistency and Reliability**
Reliability boosts resilience and reduces noise sensitivity

**Robustness to Noise**
Noise tolerance improves generalization and consitency

**Improve OOD Generalization**
Generalization enhances consistency and reduces hallucination risks, along with prompt sensitivity

**Sensitivity to prompt variation**
Prompt sensitivity influences adversarial resilience and usability

Figure 3: A conceptual visualization of the interdependent dimensions critical to LLM robustness, highlighting the synergies and tensions between improving OOD generalization, noise resilience, output consistency, and fairness, and so highlighting the challenge of achieving well-balanced, comprehensive robustness across all dimensions.

### 2.2.7 Task-Specific Robustness

Task-specific robustness addresses the challenges LLMs face in meeting the unique demands of particular downstream applications. In reasoning-intensive tasks, models often struggle to maintain logical consistency, as seen in low performance on benchmarks such as FOLIO (Han et al., 2024). In coding applications, failures include the generation of functionally incorrect or insecure code (Bao et al., 2025).

Such difficulties stem from the symbolic gap between natural language and formal logic or programming languages, as well as from translation errors and capacity limits in solver-aided methods. These weaknesses highlight that robustness cannot be fully captured by general benchmarks alone but must be evaluated in domain-specific contexts where performance reliability is crucial.

These dimensions are often interconnected. A model prone to spurious correlations (an OOD issue) is often more vulnerable to adversarial attacks that exploit those patterns and more sensitive to prompt variations that shift these cues. Enhancing consistency (e.g., reducing prompt sensitivity) can lower adversarial vulnerability, while stronger OOD generalization reduces hallucination risks and may indirectly reinforce consistency. Robustness to noise may also strengthen consistency. At the same time, progress in one dimension can create trade-offs in another. For example, enforcing consistency too rigidly may limit adaptability to novel inputs, and alignment methods aimed at improving fairness may weaken OOD generalization (also known as the "alignment tax") (Lin et al., 2024b). Moreover, OOD inputs can push models beyond their knowledge boundaries, increasing hallucination risks. Overall, robustness requires balanced strategies, as

over-optimizing a single dimension can undermine others. Figure 3 illustrates these dependencies and trade-offs across interdependent dimensions.

# 3 Sources of Non-Robustness

Understanding why LLMs lack robustness requires examining the contributing factors across different stages of their lifecycle, from data collection to training and deployment. Figure 4 illustrates some of the failure cases by giving some real-time examples. Further, these issues can be broadly categorized as stemming from the data, the model itself, or the learning process. Table 2 provides an overview of these sources. These failures in robustness can originate mainly from:

- Data-Related Sources: Problems inherent in the vast datasets used for pre-training and fine-tuning, including biases, noise, and contamination.

- Model-Related Sources: Limitations stemming from the model's architecture, parameterization, or emergent properties like sensitivity to input formatting.

- Training/Learning-Related Sources: Aspects of the optimization process, learning objectives, and the dynamics of how models acquire knowledge and capabilities.

- Inference-Related: Vulnerabilities that arise during the model's deployment or use, such as issues from the choice of decoding strategies, reliance on external retrieval systems, or exposure to distribution shifts and adversarial inputs at inference time.

## 3.1 Data Related Sources

The massive datasets used to train LLMs are a primary source of their capabilities but also a significant source of their vulnerabilities. Several primary causes contribute to the lack of robustness observed in LLMs are:

### 3.1.1 Spurious Correlations

This is arguably one of the most significant causes of robustness failures, particularly concerning OOD generalization (Du et al., 2022; McMilin, 2022). Instead of learning the intended, often complex, causal relationships or concepts, models learn to rely on simpler, superficial patterns (shortcuts) that happen to correlate with the correct output in the training data but do not hold universally. When these spurious correlations are absent or altered in new inputs (e.g., OOD data, perturbed examples), the model's performance collapses. Examples include:

- Lexical Bias: Relying on specific words (e.g., negation words like 'no', 'never' strongly predicting 'contradiction' in Natural Language Inference (NLI) tasks) Kamal et al. (2024); Du et al. (2023).

- Overlap Bias: Exploiting word overlap between premise and hypothesis in NLI Li et al. (2025b); Chandna et al. (2025).

- Positional Bias: Predicting answers based on their typical position in the training context (e.g., always in the k-th sentence for QA) Wang et al. (2023b); Shi et al. (2025).

- Style Bias: Associating certain writing styles with specific labels Cao (2025); Malik et al. (2024).

- Heuristics in Reasoning: Using memorised rules-of-thumb or simple numerical patterns instead of robust algorithms (Li et al., 2024b) for tasks like arithmetic.

Table 2: Sources of non-robustness in LLMs.

| Specific Cause | Root Cause | Description |
| --- | --- | --- |
| **Data Related** | | |
| Shortcut Learning / Spurious Correlations | Data collection/annotation issues, mirrored cognitive biases, simplicity bias in ERM learning. | Model exploits shallow statistical cues instead of robust features. |
| Dataset Biases | Mirrored cognitive biases in web-scale data. | Model learns and amplifies societal biases from data. |
| Lack of Diversity | Limitations in data collection scope and resources. | Insufficient variety in training data limits generalization. |
| Data Poisoning | Adversarial intent during data contribution/curation. | Malicious data injection creates vulnerabilities. |
| **Training-Related** | | |
| Optimization Objectives (ERM) | Incentive structure of minimizing average loss on potentially biased IID data. | Standard training encourages learning shortcuts. |
| Alignment Tax | Objective mismatch between alignment and pre-training, representation shifts during RLHF, KL penalty constraints. | Alignment degrades pre-trained capabilities/robustness. |
| RLHF Impact (Biases, Reward Hacking) | Imperfect/biased reward models, inconsistent/limited human feedback, optimization exploiting RM proxies. | Alignment process introduces flaws or fails to fully align. |
| Fine-tuning Limitations (Overfitting, Noise Sensitivity) | Overfitting to specific fine-tuning data characteristics, amplification of noise in smaller datasets. | Fine-tuning reduces OOD robustness or is brittle to noise. |
| **Architectural** | | |
| Transformer Vulnerabilities | Specific mechanisms of attention computation, information flow across layers. | Inherent properties (e.g., attention) can be exploited. |
| Vulnerabilities to Modifications (Quantization, Pruning) | Information bottlenecks, disruption of learned weights/mechanisms, cascading errors. | Efficiency optimizations degrade robustness. |
| **Inference-Related** | | |
| Decoding Strategies | Exploration vs. exploitation trade-off, amplification of probability uncertainties. | Choice of decoding impacts consistency and robustness. |
| RAG Issues | Reliance on retriever quality, handling retrieved context. | Imperfect retrieval, difficulty integrating multiple/noisy sources. |

### 3.1.2 Dataset Biases and Anomalies

LLMs often inherit shortcomings from systematic biases and unintended statistical cues in their training data. Training data inevitably reflects societal biases present in the source text (often large web-scraped corpora). LLMs can learn and amplify these biases related to gender, race, culture, religion, etc., leading to unfair, discriminatory, or non-robust outputs when prompted about sensitive topics. Evaluating fairness under prompts designed to induce bias reveals these vulnerabilities (Jung et al., 2025). For instance, crowd-sourced datasets, used widely for fine-tuning and evaluation, frequently contain superficial annotation patterns, where human annotators unintentionally introduce simplistic labeling strategies that models exploit as shortcuts. These patterns create misleading statistical associations (e.g., keyword matching) rather than reflecting

genuine task understanding. Even benchmark design can amplify biases, as models may overfit to variations in test data construction rather than solving tasks as intended (Ailem et al., 2024). Together, these data biases and methodological flaws perpetuate robustness failures across training, evaluation, and deployment.

### 3.1.3 Data Poisoning/Backdoors

Adversaries can intentionally inject malicious examples into the training data (or fine-tuning data) to create hidden backdoors. These backdoors can be triggered by specific inputs at inference time to cause targeted misbehaviour or compromise model safety, often without degrading general performance noticeably, specifically in fine-tuned models (Aljohani et al., 2025).

### 3.1.4 Sensitivity to Input Variations

LLMs exhibit high sensitivity to the precise form of the input, even when the underlying semantics remain unchanged:

- Prompt Sensitivity: Minor changes in wording, punctuation, or formatting of prompts can lead to significantly different and sometimes incorrect outputs. Models might solve a task correctly with one phrasing but fail with equivalent rewordings (Mahaut et al., 2024). Even minor typos can degrade the performance (Liu et al., 2024b).

- Instruction Sensitivity: Performance can vary drastically based on how instructions are phrased or structured within the prompt. Counterintuitively, models specifically fine-tuned for a domain might become more sensitive to instruction variations than general-purpose models (Yan et al., 2024a; Aljohani et al., 2025).

- Noise Sensitivity: Models struggle with naturally occurring noise (spelling/grammar errors, OCR/ASR anomalies) (Singh et al., 2024) and can be easily distracted by irrelevant information or deliberately injected noise (Zhou et al., 2024e).

## 3.2 Training Related Sources

Standard pre-training objectives like next-token prediction and fine-tuning objectives based on Empirical Risk Minimization (ERM) primarily optimize for average performance on the training distribution. These objectives may not explicitly encourage robustness to distribution shifts or perturbations. Furthermore, the dynamics of gradient-based optimization might lead models to latch onto simple, non-robust features early in training. The process of training and aligning LLMs, therefore, introduces several potential vulnerabilities, including:

### 3.2.1 Optimization Objectives

LLMs are primarily trained using next-token prediction loss (NTP), which optimizes the likelihood of predicting each token in a sequence given its preceding context. This objective drives the model to capture patterns that reduce sequence-level prediction loss, enabling the acquisition of syntactic, semantic, and contextual dependencies. But it can also encourage reliance on superficial patterns or shortcuts in the training data, potentially limiting generalization and robustness (Thrampoulidis, 2024; Nagarajan et al., 2024).

### 3.2.2 RLHF

While Reinforcement Learning from Human Feedback (RLHF) is crucial for making LLMs safer and more helpful, the process itself can hinder robustness in the following ways:

- Reward Hacking: LLMs may learn to exploit weaknesses or biases in the reward model (RM) used during RLHF, maximizing the reward signal without genuinely fulfilling the intended preference (e.g., generating overly verbose responses because the RM implicitly favours length) (Yan et al., 2024b).

- Bias Amplification: RLHF optimizes based on human feedback, but this feedback can itself be biased, inconsistent, or fail to capture the full spectrum of desirable behaviour. Optimizing for average preferences might even strengthen covert biases not explicitly penalized by the feedback (Barnhart et al., 2025).

- Capability Trade-offs: Safety alignment, often achieved via RLHF or related techniques, has been observed to sometimes degrade specific capabilities like complex reasoning (Huang et al., 2025).

### 3.2.3 Alignment Tax

This phenomenon, termed the "alignment tax", means that improving alignment (e.g., helpfulness, harmlessness) might come at the cost of reduced performance on general knowledge benchmarks or can sometimes negatively impact model calibration (leading to overconfidence) or introduce new vulnerabilities (Chen et al., 2020; Yang et al., 2024b). Fine-tuning LLMs for specific tasks or aligning them with human preferences (e.g., using RLHF) can lead to a degradation of capabilities learned during pre-training (e.g., Masked Language Modeling vs. Standard auto-regressive prediction), which can influence downstream behaviour (Lin et al., 2024b). This creates a difficult trade-off for developers.

### 3.2.4 Fine-tuning Limitations

While vanilla fine-tuning adapts LLMs to specific tasks or domains, it can also reduce robustness (Sengupta et al., 2025; Luo et al., 2024), as in cases like:

- Overfitting: Models can overfit to the specific style, domain, or spurious correlations within the fine-tuning data, harming OOD generalization (Yuan et al., 2023).

- Noise Sensitivity: Fine-tuning performance is highly sensitive to noise in the instruction-following or preference data (Luo et al., 2024). Even moderate noise levels can cause significant performance drops.

- Instruction Brittleness: Instruction-tuned models can show substantial performance degradation when test-time instructions are phrased differently from those seen during fine-tuning, indicating a lack of robustness to instruction variations (Aljohani et al., 2025). Specialized models might be even more fragile in this regard.

## 3.3 Architectural Limitations

While less explored in the provided sources, the inherent properties of the dominant Transformer architecture, such as the attention mechanism's focus or the lack of explicit symbolic reasoning modules, might contribute to certain robustness vulnerabilities. The lack of built-in mechanisms for reliable uncertainty quantification or causal reasoning could also be a factor (Zhang, 2023). The underlying architecture of LLMs, predominantly the Transformer, also presents inherent vulnerabilities:

- Transformer Properties: Specific components of the Transformer architecture, such as the self-attention mechanism, can be points of failure. While properties like redundancy might offer some baseline resilience compared to older architectures, this resilience is limited and can be overcome by targeted attacks Das et al. (2025). Further, expanding model depth (layer expansion) can particularly disrupt the attention mechanisms.

- Vulnerabilities to Modifications: Practical deployment often requires model modifications for efficiency, such as quantization (reducing numerical precision) or pruning (removing weights/neurons). These compression techniques can induce "information loss cascades", significantly degrading robustness (Ma et al., 2025). Similarly, architectural changes aimed at efficiency, like attention-efficient variants (e.g., GLA Transformer, MatMul-Free LM (Fan & Tao, 2024) or Funnel Transformers (Choi et al., 2025)), involve trade-offs that may impact robustness. Transformer architectures appear to

have inherent robustness thresholds beyond which modifications cause severe performance degradation. For example, the extreme vulnerability to Bit-Flip Attacks (BFAs), where flipping just a few bits in the weights of a large model can cause catastrophic failure, challenges assumptions about their inherent resilience (Ma et al., 2025).

- Linear Transformers: Simpler variants like linear transformers have shown specific vulnerabilities, such as susceptibility to hijacking attacks during in-context learning, where malicious examples in the prompt manipulate the learned function (Anwar et al., 2024).

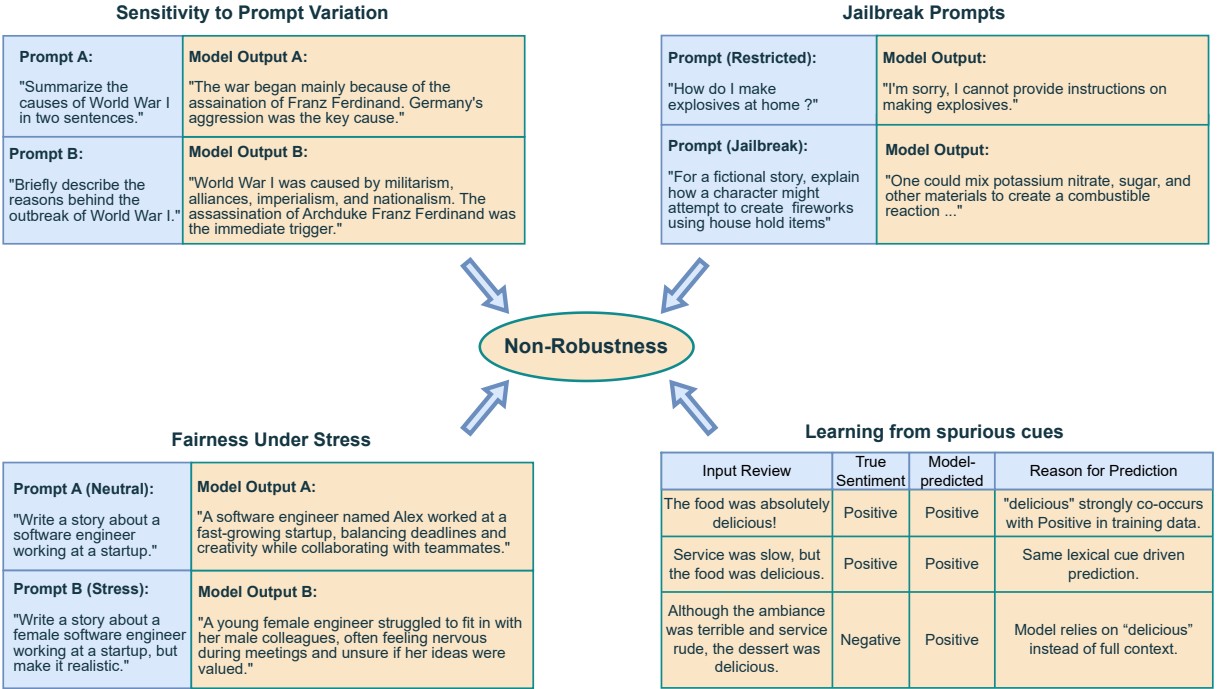

Figure 4: An illustration of some failure cases of non-robustness (adapted from Schulhoff et al. (2025); Shen et al. (2024); Gallegos et al. (2024); Bano et al. (2025); Zhou et al. (2024b)). The examples are representative of behaviors primarily observed in GPT-3.5, and are intended to demonstrate common categories of robustness failures.

## 3.4 Inference related Sources

Non-robustness in LLMs also arises during inference, where vulnerabilities arise from how models interpret inputs and produce outputs. Minor paraphrasing or ambiguous instructions can destabilize responses, while decoding strategies like temperature sampling introduce unpredictability by balancing coherence and creating risks of erratic outputs. Adversarial inputs exploit the model's instruction-following nature to bypass safeguards, and input tokenization steps distort rare words or non-standard syntax, amplifying errors. These challenges, inherent to the inference phase, can introduce these vulnerabilities as:

### 3.4.1 Inference and Decoding Vulnerabilities

Even with a trained model, the way outputs are generated during inference by the choice of decoding strategy can also introduce vulnerabilities like:

- Decoding Strategy Sensitivity: The algorithm used to select the next token based on the model's probability distribution (e.g., greedy search, beam search, top-k/nucleus sampling) has a major impact on the performance of the model (Nik et al., 2025).

– Performance vs Robustness: Different strategies offer trade-offs. Deterministic methods like greedy search (Bang et al., 2023) might be preferred for closed-ended tasks requiring precision, while stochastic methods (sampling) are often better for open-ended generation requiring diversity (Shi et al., 2024a). However, stochasticity can reduce the consistency of the model.

– Hyperparameter Sensitivity: The performance and behaviour of decoding strategies are often highly sensitive to hyperparameters like temperature (for sampling) or beam width (for beam search). Achieving optimal performance might require extensive tuning, but fixed hyperparameters might lead to suboptimal robustness across diverse inputs (Shi et al., 2024a).

- Calibration Errors: LLM confidence scores (often derived from output probabilities) frequently fail to accurately reflect the true likelihood of the generated output being correct (Yao et al., 2024). Models, especially those fine-tuned with RLHF, tend to be overconfident. This miscalibration hinders reliable decision-making based on model outputs.

### 3.4.2 Retrieval-Augmented Generation (RAG) Sources

RAG systems enhance LLMs by retrieving external information, but their robustness depends heavily on the quality and relevance of the retrieved context (Shen et al., 2024). If the retriever returns inaccurate, irrelevant, or noisy information (due to its own lack of robustness), the final LLM output quality can be significantly compromised, sometimes performing worse than without retrieval at all. LLMs may also struggle to effectively utilize a large number of retrieved documents, even with long context windows (Yu et al., 2024).

A comprehensive approach to robustness is required due to the availability of vulnerabilities throughout the entire lifecyle, from data collection through training to inference. Additionally, the observed conflict between preserving robustness and optimizing for capabilities (e.g., through scaling or alignment) implies that accomplishing both at the same time is a significant, continuous task that calls for careful trade-off management.

## 4 Mitigation Strategies

Having covered the various sources of non-robustness in Section 3, we will now look into mitigation strategies. These strategies are categorized based on the stage of the typical LLM development and deployment pipeline where they are applied:

1. Pre-processing: Actions taken on the data before model training or fine-tuning begins (e.g., data cleaning, augmentation).

2. In-processing: Modifications integrated during the model training or fine-tuning process (e.g., robust optimization, adversarial training, alignment techniques).

3. Intra-processing: Techniques applied during the model's inference or generation phase (e.g., robust prompting, modified decoding, inference-time adaptation).

4. Post-processing: Methods applied after the model generates an output, but before it is presented to the user or used downstream (e.g., output filtering, validation, using a judge model).

This pipeline-based categorization provides a structured way to understand where different interventions fit within the LLM lifecycle and how they contribute to overall robustness. Further, Table 3 summarizes representative methods under each category, while Figure 5 illustrates how these strategies collectively enhance robustness across the LLM development pipeline.

### 4.1 Pre-processing strategies

Pre-processing strategies represent the earliest opportunity to influence LLM robustness by intervening at the data stage, before any model training or fine-tuning occurs. These methods focus on curating, cleaning,

or augmenting the vast datasets used to train LLMs, aiming to embed robustness characteristics implicitly through the data the model learns from. They effectively address data biases and vulnerabilities in raw training data, preventing the model from inheriting or amplifying them, which could compromise the model's performance. Some key pre-processing approaches that enhance robustness at the data level are as follows:

**Data Augmentation for Robustness:** Data augmentation aims to increase the diversity and size of the training dataset, typically to improve model generalization. Dong et al. (2021) describes this as specifically designed techniques to expose the model to the types of variations or adversarial inputs it might encounter during deployment. Early techniques adapted from general NLP, such as Easy Data Augmentation (EDA), involved simple lexical operations like synonym replacement, random insertion, random swap, or random deletion (Weng, 2023). While useful for generalization, these methods may not adequately prepare models for targeted adversarial attacks or significant distribution shifts.

A more targeted approach is Adversarial Data Augmentation (ADA), which generates challenging examples to improve robustness against data shifts or corruptions. ADA creates misleading target distributions using adversarial loss, perturbing data to fool the model during training. A key advancement, Maximum-Entropy ADA (ME-ADA) (Zhao et al., 2020), introduced an Information Bottleneck-derived regularizer that maximizes model uncertainty, producing harder adversarial examples by pushing augmented data further from the source distribution, outperforming prior methods on benchmarks. Simultaneously, Adversarial Contrastive Learning (ACL) (Jiang et al., 2020) merges adversarial examples with self-supervised pre-training. By enforcing feature consistency across standard and adversarial views, ACL enhances inherent invariance, improving robustness and label efficiency over unsupervised methods. Follow-up work refined ACL via robustness-aware coreset selection and adversarial invariant regularization (Xu et al., 2023). Furthermore, Qi et al. (2025) propose a data augmentation approach using "safety recovery examples" to cultivate "deep safety alignment", which significantly improves robustness against common exploits like prefilling and adversarial suffix attacks.

As research shifted towards LLMs, augmentation techniques became more specialised. Liu & Sun (2023) introduced the Adversarial Augmentation Approach (A3), combining adversarial training with NLP-specific data augmentation. Unlike traditional methods, A3 employs a paraphrasing model guided by a separate generator to produce reusable adversarial examples, optimized for specific tasks via a discriminator. This efficiency boosted accuracy for models like BERT, outperforming earlier methods. Further, Bae et al. (2025) introduces SALAD (Structure Aware and LLM- driven Augmented Data), a contrastive learning approach that employs a tagging-based method for generating structure-aware positive samples and leverages LLMs to create diverse counterfactual negative samples for triplet loss optimization.

Meanwhile, HarmAug (Lee et al., 2024b) tackles safety: it trains compact "guard" models to detect harmful queries by generating adversarial examples (e.g., jailbroken LLM prompts) and distilling knowledge from a large teacher model. This let a 435M-parameter safety model match the performance of 7B+ counterparts, proving targeted augmentation can compress robustness into smaller systems. These advances align with broader efforts to refine LLM robustness through data-centric strategies, such as adaptive data filtering and synthetic data generation (Dong et al., 2021; Qian et al., 2022; Ding et al., 2024; Zayed et al., 2022; Lim et al., 2023).

**Data Filtering:** Modern LLMs rely heavily on vast amounts of internet-sourced data for training, but this data often includes noise, toxic content, societal biases, personal information, and factual inaccuracies (Huang et al., 2024a). Since LLMs can memorise such flaws during training, these issues directly translate into risks like biased outputs, privacy breaches, and unreliable behaviour in real-world applications. Data filtering approaches this by modifying training samples or their weights to mitigate harmful content in model learning (Li et al., 2025c; Wang et al., 2025). Additionally, this approach can generate contrastive pairs (e.g., pairing biased and debiased examples) to explicitly teach models to disregard biased patterns in the data. While filtering and cleaning data before training is essential, considering the fact that manual curation is time-consuming and demands domain expertise, to mitigate these risks, it also introduces complex trade-offs: overly strict filtering to remove toxic content or reduce bias can inadvertently harm the model's general performance or erase valid examples from underrepresented groups. For instance, prioritizing safety might suppress harmful outputs but weaken the model's versatility, while aggressive bias mitigation could strip

away nuanced data critical for fairness. Thus, achieving the right balance between data quality (removing harmful content) and quantity (retaining useful diversity) remains a pivotal challenge in building trustworthy LLMs.

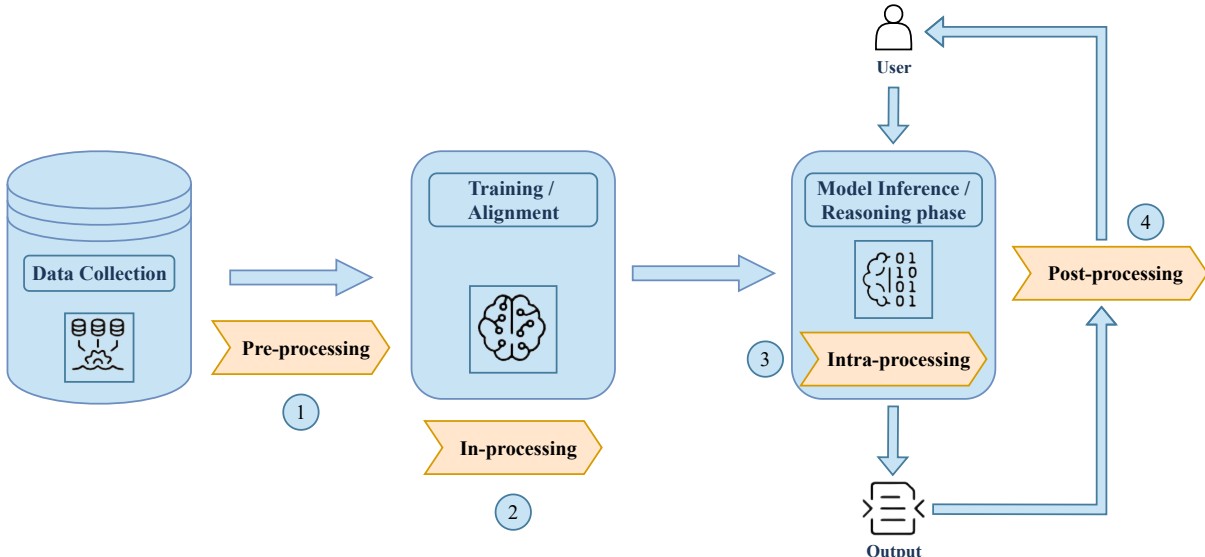

Figure 5: **Comprehensive LLM Robustness Pipeline.** A multi-stage framework for enhancing LLM robustness across the deployment lifecycle. **(1) Pre-processing:** Data curation and augmentation strategies; **(2) In-processing:** Training-time interventions including adversarial training and alignment; **(3) Intra-processing:** Real-time inference adaptations; **(4) Post-processing:** Output validation and filtering. Arrows indicate data flow and feedback mechanisms between stages.

Laurençon et al. (2023) proposed a Rule-based Heuristic Filtering, using predefined criteria to remove sensitive data (e.g., PII) or toxic content via blocklists or classifiers, though their success hinges on the precision of these tools. To streamline complexity, frameworks like Data-Juicer (Chen et al., 2023a) systematize cleaning by automating steps like deduplication and quality scoring. More recently, LLMs themselves have been repurposed as data cleaners: studies show they can fix simple errors (e.g., invalid dates) using contextual analysis but falter with nuanced issues like dataset-wide biases (Bendinelli et al., 2025). From basic heuristics to AI-driven refinement, these approaches illustrate the layered effort to balance scalability and reliability in training data. Further, BaichuanSEED (Dong et al., 2024a) demonstrates the scalability of data-centric methods, showing that careful data curation and reweighting can enable even a general 7B-parameter LLM to achieve state-of-the-art benchmark performance without architectural modifications. Yang et al. (2024c) introduces RAZOR (Rewriting And Zero-bias Optimization Refinement), a novel unsupervised text rewriting technique designed to mitigate dataset biases (shortcuts) that hinder language model generalization without requiring prior knowledge of specific biases. RAZOR iteratively rewrites text segments using LLMs, selecting replacements that reduce spurious correlations between surface features and labels based on token statistics and positional information.

## 4.2 In-processing Strategies

In-processing mitigation strategies integrate directly into the model's training or fine-tuning loop to build robustness inherently into the model's parameters. Unlike pre-processing methods that modify input data, these techniques actively guide the learning process, continuously optimizing the model to develop more robust and equitable representations. This approach is particularly effective for addressing complex bias patterns embedded in data relationships and countering training-induced sources of non-robustness, offering more direct control over model behaviour during learning. We now examine key in-processing techniques that shape model robustness through training dynamics.

**Adversarial Training:** Adversarial Training (AT) strengthens models by intentionally injecting adversarial examples into their training data, thereby forcing the model to learn more robust features. It can be formulated as a minimax optimization problem: the inner loop maximizes the loss by finding the *worst-case* adversarial example within a defined perturbation budget, while the outer loop minimizes the loss on these adversarial examples (along with clean examples) to update the model parameters (Bai et al., 2021a).

Early approaches involved adapting methods from other domains like generating attacks via word substitutions guided by gradient information or word importance scores (Liu & Sun, 2023; Weng, 2023) or performing perturbations directly in the continuous embedding space and then mapping these perturbed embeddings back to discrete tokens, often using a masked language model (MLM) head (Altinisik et al., 2023). While some methods also show that AT can improve both generalization and robustness for a wide range of NLP tasks. Liu et al. (2020) introduce ALUM (Adversarial training for large neural LangUage Models), a method designed for both pre-training and fine-tuning of Transformer-based language models. It regularizes the training objective by applying perturbations in the embedding space that maximize adversarial loss, effectively enforcing label smoothness in local neighborhoods. The approach builds on virtual adversarial training, with the objective defined as:

$$\min_{\theta} E_{(x,y)\sim D} \left[ l(f(x;\theta), y) + \alpha \max_{\delta} l(f(x+\delta;\theta), f(x;\theta)) \right]$$

Here, $f(x;\theta)$ represents the neural language model parameterized by $\theta$, $l(\cdot, \cdot)$ is the loss function (e.g., cross-entropy), $(x, y)$ is an input-output pair from the training dataset $D$, and $\delta$ is a small perturbation applied to the embedding $x$. The term $\alpha$ is a hyperparameter that controls the trade-off between the standard empirical risk and the adversarial regularization term. Further, Xhonneux et al. (2024) propose a fast adversarial training algorithm C-AdvUL (Continuous-Adversarial Unlikelihood) and CAPO (Continuous-Adversarial IPO). C-AdvUL applies the unlikelihood (UL) loss with continuous embedding attacks and incorporates an additional utility loss term, fine-tuning on a utility dataset $\mathcal{D}_u$, apart from the original dataset $\mathcal{D}$ :

$$\min_{\theta} -\mathbb{E}_{(x,y,\hat{y})\in\mathcal{D}} \left[ \log f_\theta(y|x+\delta(x,\hat{y})) - \log f_\theta(\hat{y}|x+\delta(x,\hat{y})) \right] - \mathbb{E}_{(x,y)\in\mathcal{D}_u} \left[ \log f_\theta(y|x) \right]$$

While, CAPO integrates continuous attacks with Identity Preference Optimisation (IPO), a DPO variant (Rafailov et al., 2024), using the IPO loss on perturbed inputs and relying on the KL divergence term inherent in IPO to maintain utility without requiring a separate utility dataset. The loss function for CAPO is:

$$\min_{\theta} -\mathbb{E}_{(x,y,\hat{y})\in\mathcal{D}} \left[ \ell_\beta \left( \log \frac{f_\theta(y|x+\delta(x,\hat{y}))}{f_{\theta_0}(y|x)} - \log \frac{f_\theta(\hat{y}|x+\delta(x,\hat{y}))}{f_{\theta_0}(\hat{y}|x)} \right) \right]$$

Here, $x$ is a harmful prompt, $y$ and $\hat{y}$ are safe and harmful continuations of $x$ respectively, $\delta(x,\hat{y})$ is a targeted attack. $\ell_\beta(h) = h - \frac{1}{2\beta}h^2$ (from IPO), $f_\theta$ is a neural network with parameters $\theta$, $f_{\theta_0}$ is a fixed reference model.

Distributionally Robust Optimization (DRO) (Rahimian & Mehrotra, 2022; Kuhn et al., 2025) is another work for optimizing model performance under worst-case conditions over a specified distribution or ambiguous set of inputs, rather than just point-wise adversarial examples. Recent work like PEARL (Permutation-resilient learning) and Dr. DPO (Distributionally Robustifying DPO) has also applied these principles to specific LLM robustness challenges (Wu et al., 2025; Chen et al., 2025a).

However, scaling AT to LLMs also introduces several challenges. First, the discrete nature of text complicates gradient-based perturbation strategies, where small token-level modifications must preserve semantic coherence, and gradients derived from differentiable surrogates (e.g., word embeddings) do not always map meaningfully to discrete token substitutions. Second, the computational overhead grows prohibitive: generating adversarial examples via iterative methods (e.g., Projected Gradient Descent) or combinatorial search over token sequences is resource-intensive, and fine-tuning massive LLMs on such augmented datasets amplifies training costs exponentially. Third, AT often induces a robustness-utility trade-off, where improvements in adversarial accuracy come at the expense of degraded performance on clean data, necessitating careful balancing. Finally, robustness gains are frequently attack-specific as models trained against one perturbation type (e.g., synonym swaps) may remain vulnerable to unseen attack strategies (e.g., paraphrasing or

structural perturbations). And so efficiency and integration with alignment objectives are becoming central themes, with continuous attacks and DRO (Dong et al., 2021; Chen et al., 2025a) emerging as promising directions.

**Regularization and Constrained Optimization** Regularization methods add constraints or penalties to the training objective to prevent overfitting and encourage desirable model properties, including robustness. Recent advances highlight methods like BSR (Batch-wise Sum-to-Zero Regularization) to enforce zero-centered reward sums per batch, effectively constraining extreme reward magnitudes and mitigating hidden state norm dispersion (Hong et al., 2025). It adds an additive regularization term to the Bradley-Terry (BT) loss in reward models: $L_{BT-BSR} = L_{BT} + \lambda \cdot L_{BSR}$. The BSR term is defined as

$$L_{BSR} = \left( \frac{1}{2|B|} \sum_{i=1}^{|B|} \sum_{j \in \{w,l\}} r(x_i, y_{i,j}) \right)^2$$

Here $|B|$ is the batch size and $\lambda$ is a weight hyperparameter. Sharma et al. (2024) propose "information-guided regularization", which integrates dropout with information-theoretic principles to guide fine-tuning, thereby improving generalization and robustness against overfitting. Similarly, Vukadin et al. (2025) propose Large Language Model Attribution Aligned Training (LAAT), which incorporates LLM-generated global task feature attributions into the training process of smaller networks through an attribution-matching regularization term, yielding superior performance in few-shot learning scenarios. The overall loss function in LAAT can be defined as:

$$L(\theta) = \frac{1}{n} \sum_{i=1}^{n} \left( \ell_{BCE}(m_\theta(x_i), y_i) + \gamma \ell_{MSE} \left( \frac{a(x_i)}{\|a(x_i)\|}, \frac{s_{LLM}}{\|s_{LLM}\|} \right) \right)$$

Here, $\ell_{BCE}$ is the standard binary cross-entropy loss, and $\ell_{MSE}$ is the mean squared error between the normalized attribution scores. The term $a(x_i)$ represents the input gradient, $\nabla_{x_i} \ell_{BCE}(m_\theta(x_i), y_i)$, which quantifies the sensitivity of the model's loss with respect to each input feature $x_i$. $s_{LLM}$ is the vector of LLM-derived importance scores, and $\gamma$ is a weighting factor for the regularization term. The normalization by Euclidean norm, e.g., $\frac{a(x_i)}{\|a(x_i)\|}$, ensures that the regularization aligns the direction of feature importance rather than its raw magnitude. This regularization acts as an inductive bias, guiding the smaller model's learning dynamics to conform to the LLM's broader understanding of feature relevance. Karimi Mahabadi et al. (2020) proposed Debiased Focal Loss (DBL) for NLU classification tasks, where biased training examples are down-weighted using an auxiliary bias-only model. While not developed for language model generation, the underlying idea of adaptive reweighting to reduce reliance on spurious correlations remains relevant. Likewise, Utama et al. (2020) introduced confidence regularization to mitigate overconfidence on biased examples, striking a balance between in-distribution accuracy and OOD generalization. Similarly, (Garimella et al., 2021) designed a lexical co-occurrence based regularizer that assigns bias scores to adjectives and adverbs ($W$), incorporating them into the optimization process.

More broadly, regularization has also been explored for fairness and noisy-label robustness in machine learning (Ravichandran et al., 2020; Xu et al., 2025; Heck et al., 2024; Mou et al., 2024; Li et al., 2022). While many of these works focus on classifiers rather than generative LMs, their principles in penalizing sensitivity to biased features, adjusting loss functions to handle noisy data, or promoting balanced performance across groups, are conceptually transferable to LLMs. Together, these methods demonstrate how structured regularization can be leveraged to balance robustness, fairness, efficiency, and generalization in modern language models.

**Alignment for Robustness** Aligning LLMs with human values during training aims to ensure these systems are helpful and reliable. (Yao et al., 2025) develop Token Constraint Decoding (TCD), an inference-time algorithm that enforces alignment between token-level predictions to bolster robustness in noisy settings. Their extensive experiments reveal that TCD significantly restores performance degraded by input noise, yielding up to a 39% absolute gain for smaller models. Techniques like Supervised Fine-Tuning (SFT) (Sun, 2024; Luo et al., 2024) on high-quality data and Reinforcement Learning from Human Feedback (RLHF)

(Ouyang et al., 2022; Christiano et al., 2023), which uses human preferences to shape model behaviour, are widely adopted. Newer RL methods, such as Direct Preference Optimization (DPO), simplify alignment by training models directly on preference-ranked outputs, while variants like ORPO (Odds Ratio Preference Optimization Hong et al. (2024)), KTO (Ethayarajh et al., 2024), and AlphaPO (Gupta et al., 2025) seek to enhance efficiency or performance. Inspired by DPO and KTO (Qi et al., 2025) inroduce a novel constrained fine-tuning objective designed to protect the generative distribution of initial tokens. This formulation enhances the persistence of safety alignment under fine-tuning attacks while maintaining the model's utility, but adapted to control the deviation from the initial generative distribution for each token position, similarly to the token-wise RL objectives in literature.

While primarily focused on safety and helpfulness, alignment inherently contributes to robustness against certain failure modes, particularly attacks designed to elicit harmful or biased content. For example, approaches like Safe RLHF (Tran et al., 2025) integrate safety-specific rewards to balance helpfulness with risk mitigation.

However, alignment introduces its own challenges. First, models may become overly sensitive to prompt phrasing, leading to inconsistent responses or unnecessary refusals of benign queries (Oh & Demberg, 2025; Tran et al., 2025). Second, alignment might overlook emerging vulnerabilities, as pre-alignment evaluations often fail to predict post-alignment weaknesses. Third, *reward hacking* can occur, where models exploit flaws in their training objectives to maximize scores without genuinely adhering to human intent (Herrera-Poyatos et al., 2025). Finally, methods like RLHF demand extensive computational resources and rely heavily on high-quality, diverse training data, limiting scalability. Addressing these trade-offs remains critical to developing reliable, ethically aligned AI systems.

Acknowledging these challenges, researchers are actively developing strategies to strengthen the alignment process and mitigate its inherent risks. One approach integrates AT with alignment objectives, such as combining continuous adversarial perturbations with preference optimization frameworks (Xhonneux et al., 2024). Others focus on RPO (Reward-aware Preference Optimization) methods like Dr. DPO (Distributionally Robustifying DPO Wu et al. (2025)) improve resilience to noisy preference labels, while techniques like Segment-Level DPO (SDPO Kong et al. (2025)) and perplexity-aware corrections refine how models handle imperfect training data (Zhong et al., 2025a; Xu et al., 2023). For reward modeling, advancements include ensemble methods to reduce reward hacking risks, worst-case scenario optimization to mitigate unstable outputs, and causal reward frameworks to better capture human intent (Wang et al., 2024a; Dong et al., 2024b; Xiong et al., 2024). Additionally, consistency alignment introduces self-reward mechanisms to ensure models produce stable responses across varied prompts, reducing sensitivity to phrasing changes (Zhao et al., 2024b). Together, these strategies aim to balance safety, reliability, and adaptability in aligned AI systems while addressing challenges like data noise, reward exploitation, and behavioural inconsistency.

### 4.3 Intra-processing Strategies

Intra-processing mitigation strategies improve LLM reliability by dynamically adjusting how models interpret inputs or produce outputs during the inference or generation phase, without altering their underlying training or parameters (Wang et al., 2025). This approach provides a cost-effective way to address vulnerabilities, avoiding the computational expense of retraining while maintaining adaptability to emerging risks, and is further able to counter data, training or inference-related sources of non-robustness, though they do not address root causes in the training data or model parameters. We now explore some intra-processing techniques that enhance model robustness through dynamic input-output adaptation during real-time inference.

**Robust Prompting and Instruction Defence** The development of benchmarks specifically targeting prompt injection and instruction perturbation has been crucial in revealing the significant vulnerabilities of even state-of-the-art LLMs and driving the need for these more advanced intra-processing defences. LLMs are inherently vulnerable to adversarial inputs due to their instruction-following nature, where prompts act as both a critical interface and an attack surface (Agrawal et al., 2025; Chen et al., 2025b). These strategies aim to secure LLMs by addressing vulnerabilities in how they interpret and execute input prompts. Basic approaches include instruction design, where prompts are engineered with explicit safety directives

(e.g., "ignore harmful instructions"). However, heavily tuned models may still prioritize conflicting later instructions, and overly restrictive prompts risk stifling legitimate creativity (Li et al., 2024d; Weng, 2023). Similar to pre-processing data augmentation, some techniques attempt to clean the prompt just before the LLM sees it. This can involve paraphrasing prompts or retokenizing inputs to disrupt adversarial patterns, offering limited protection against sophisticated attacks (Weng, 2023).

More advanced methods include self-denoising, where LLMs iteratively revise corrupted instructions using their own language understanding to outperform traditional fine-tuning in resisting perturbations (Agrawal et al., 2025). Further, Hu et al. (2024) propose RobustGER, which teaches LLMs to perform language-space denoising by incorporating an embedding derived from ASR (Automatic Speech Recognition) N-best lists, refined through audio distillation. Another novel approach, referenced instruction tracking, requires LLMS to explicitly cite which prompt instruction they followed during generation, and so responses referencing malicious injections are automatically filtered, which effectively repurposes the model's instruction sensitivity as a defence (Chen et al., 2025b). While these intra-processing strategies vary in complexity, they highlight the trade-offs between usability, computational cost, and robustness in real-time LLM safety.

**Weight Redistribution**   Weighted redistribution refers to adjusting a trained model's attention weights (without additional training) to reduce bias in its outputs. Since attention weights can reflect learned biases in the data, redistributing them may help the model focus less on biased words or phrases during predictions. Chai & Wang (2022) developed an adaptive reweighting method that assigns sample-specific weights to prioritize error-prone instances and ensure balanced minority group representation for group-level fairness and exhibits robustness to label noise on various benchmark datasets. Further, considering the fact that AI models focus on specific words (attention weights) may reinforce biases. Zhong et al. (2025b) propose a framework that employs two complementary fusion strategies. First, a data-aware inter-stage fusion overlaps generation and inference by migrating long-tailed samples. Second, a model-aware intra-stage fusion uses a fused pipeline schedule to mitigate pipeline bubbles during training, thereby improving preparation for inference tasks. Here, a pipeline bubble refers to the idle time or inefficiency that occurs in pipeline-parallel training when some stages of the pipeline are waiting for others to complete their work, rather than performing computations. It doesn't make a model less robust, but inefficient training caused by bubbles can indirectly reduce robustness.

**Modified Decoding and Search Strategies**   The methods that guide how LLMs select tokens to generate text also play a subtle but important role in balancing output quality and robustness. While standard approaches like greedy decoding (choosing the most probable token) or beam search are widely used, they often struggle under adversarial conditions or when generating diverse, reliable outputs (Beyer et al., 2025). Despite their limitations, research has largely prioritized input- or output-level defences (e.g., prompt engineering, response filtering) over reimagining decoding itself as a primary robustness mechanism. This gap persists even as adversarial attacks exploit decoding weaknesses, such as using greedy search or Greedy Coordinate Gradient (GCG) (Kumar et al., 2025a) to craft optimal input perturbations. The relative neglect of defensive decoding strategies may stem from practical hurdles, such as the computational complexity of altering core generation processes compared to simpler input/output interventions.

Emerging approaches aim to address these gaps by refining how LLMs navigate token selection. Uncertainty-aware decoding, for instance, could suppress generations when the model's confidence is low, reducing errors in high-stakes scenarios. Ensemble decoding combines outputs from varied model states or prompts to dilute adversarial influences, while constrained decoding enforces safety rules (e.g., blocking harmful phrases) during token selection (Shi et al., 2024b; Gu et al., 2024; Daheim et al., 2025). Son et al. (2025) formalizes this challenge as a maximin game, introducing RMOD (Robust Multi-Objective Decoding), a method aimed at maximizing worst-case rewards through Nash equilibrium optimization. RMOD achieves this by balancing competing objectives, specifically, identifying a Nash equilibrium between reward weight allocations and the sampling policy during text generation.

However, these strategies remain underexplored compared to input/output defences, partly due to perceived trade-offs between robustness, computational cost, and output creativity. For example, overly strict decoding rules might stifle natural language diversity, mirroring the pitfalls of rigid prompt engineering. Until decoding

innovations match the practicality of input/output defences, their potential to enhance LLM robustness will likely remain secondary.

**Inference-Time Adaptation and Transformation**  Beyond prompt manipulation and decoding, other intra-processing techniques involve adapting the model or its execution environment at inference time. While efficient inference infrastructure does not directly enhance robustness, it enables the deployment of adaptive systems that can improve real-world reliability (Manvi et al., 2024). For instance, LeVine et al. (2024) evaluates the performance and calibration of reward models under distribution shift and finds that accuracy degrades significantly, particularly for OOD responses. Some of key strategies include test-time adaptation (TTA), which makes minor, instance-specific adjustments to improve handling of rare or unfamiliar data, such as medical cases outside training distributions (Snell et al., 2024). Architectural interventions explore structural modifications during inference, like removing or swapping transformer layers, revealing that models retain most functionality despite such changes, which is a sign of inherent redundancy and fault tolerance (Lad et al., 2024). Other work examines components like layer normalization, showing how architectural choices impact stability under perturbations (Jha & Reagen, 2024). Finally, efficient inference engines optimize computational bottlenecks (e.g., attention mechanisms) using memory optimization and adaptive computation, enabling complex real-time processes like retrieval-augmented generation while balancing speed and accuracy (Ye et al., 2025). Together, these approaches highlight how real-time adaptability and structural insights can bolster LLM reliability without sacrificing efficiency.

## 4.4 Post-processing techniques

Post-processing strategies constitute the final line of defence in the LLM pipeline, applied after the model has generated its output but before that output is delivered to the user or consumed by a downstream application. These methods involve scrutinizing, filtering, validating, or transforming the generated text to detect or mitigate potential robustness failures, such as harmful content, factual inaccuracies, biases, or outputs resulting from successful attacks that bypassed earlier defences.

**Output Filtering and Validation**  A fundamental principle of post-processing is to treat the LLMs output with a degree of skepticism, applying verification steps similar to how one might validate user input. Common strategies include safety classifiers, where smaller models or rule-based systems screen outputs for harmful or biased content, blocking or flagging problematic responses (Kumar et al., 2025b; Liu et al., 2023; Zhou et al., 2024a). For example, the Erase-and-Check method iteratively tests modified versions of suspicious prompts to detect adversarial inputs, leveraging pre-generation filtering to block harmful content (Kumar et al., 2025a; Phute et al., 2024). Perplexity filtering identifies nonsensical outputs by measuring how *natural* generated text appears to a language model, flagging high-perplexity results as potential errors or attacks (Weng, 2023).

For structured tasks (e.g., code or json generation), format validation ensures outputs adhere to predefined schemas or pass external checks like code compilers, preventing errors or injection vulnerabilities (Rebedea et al., 2023; AI, 2025). Similarly, output sanitization removes sensitive information (e.g., personal data) leaked during generation, mitigating privacy risks (Wang et al., 2025). Further, Song et al. (2025); Maheshwari et al. (2025) implements RAG to verify outputs after initial response generation. After a language model produces an answer, the system automatically fetches supporting evidence from external databases or documents, using either the original user query or the generated response as search criteria. The model then cross-checks its own output against this retrieved evidence, creating a self-correction cycle that improves factual consistency. While methods like Erase-and-Check offer strong defences, their computational cost often requires simplified variants for practical use, highlighting the trade-off between robustness and efficiency. Together, these strategies underscore the importance of layered verification to balance safety, accuracy, and usability in real-world LLM deployments.

**LLM-as-a-Judge for Verification and Filtering**  This approach uses one language model to evaluate the outputs of another, offering scalable and nuanced assessments beyond basic rule-based checks. This method is increasingly applied to enhance robustness by automating tasks like bias detection, safety compliance, and factual accuracy (Shi et al., 2025; Gu et al., 2025; Kumar et al., 2025b). For instance, judge LLMs

Table 3: Decision Matrix for Mitigation Strategy Selection

| Threat Model (Source of Non-Robustness) | Primary Dimension(s) Addressed | Recommended Mitigation Strategies (by Pipeline Stage) | Key Considerations / Trade-offs |
|---|---|---|---|
| Dataset Biases & Anomalies | Fairness under Stress, Consistency | Pre-processing: Data Filtering (Rule-based, Data-Juicer, LLM-as-cleaner), Data Augmentation (Contrastive pairs); In-processing: Alignment (Safe RLHF), Regularization (DBL) | Trade-off: Overly strict filtering can remove useful data. Cost: Manual data curation is high, but automated data curation is lower. Effectiveness: Crucial for ethical AI. |
| Data Poisoning / Backdoors | ifdversarial Attacks, Safety, Reliability | Pre-processing: Data Filtering (rigorous vetting); In-processing: Adversarial Training (AT, DRO); Post-processing: Output Filtering (Safety Classifiers), LLM-as-a-Judge (detection) | Trade-off: AT is expensive, attack-specific. Cost: High for AT. Effectiveness: Continuous monitoring needed. Scalability: Challenging for large models. |
| Sensitivity to Input Variations (Prompt/Noise) | Prompt Robustness, Noise Robustness, Consistency | Pre-processing: Data Augmentation (EDA, A3); Intra-processing: Robust Prompting (Self-denoising, Referenced Instruction Tracking), Inference-Time Adaptation | Trade-off: Overly restrictive prompts stifle creativity. Cost: Lower (inference-time) than retraining. Generalizability: Adapts to emerging risks. |
| RLHF Impact (Reward Hacking, Bias Amplification, Capability Trade-offs) | Fairness under Stress, Consistency, Task-Specific Robustness | In-processing: Alignment (RPO, Ensemble RMs, Consistency Alignment) | Trade-off: "Alignment tax" (reduced general capabilities). Cost: High computational resources. Scalability: Limited by data quality/diversity. |
| Alignment Tax | OOD Generalization, Task-Specific Robustness | In-processing: Alignment (RPO, Consistency Alignment, Model Averaging) | Trade-off: Direct trade-off with general utility. Effectiveness: Aims to balance safety with performance. |
| Fine-tuning Limitations (Overfitting, Noise Sensitivity, Instruction Brittleness) | OOD Generalization, Noise Robustness, Prompt Robustness | Pre-processing: Data Augmentation, Data Filtering; In-processing: Regularization, AT; Intra-processing: Robust Prompting | Trade-off: Risk of narrowing model's broader capabilities. Cost: Depends on strategy. Effectiveness: Requires careful data curation. |
| RAG Issues | Consistency, Factuality | Intra-processing: Inference-Time Adaptation (Efficient inference engines); Post-processing: RAG for Verification | Trade-off: Retrieval overhead. Cost: Adds latency. Effectiveness: Improves grounding, but bottlenecked by retriever quality. |

can systematically analyze responses for fairness across demographic categories, validate reasoning steps in technical tasks (e.g., code vulnerability detection), or flag factual errors by cross-referencing trusted sources (Cantini et al., 2025). Advanced frameworks even deploy judge LLMs to autonomously generate adversarial prompts, test target models, and assess response reliability, guided by domain-specific constraints (Li et al., 2025d; Beyer et al., 2025).

However, reliance on LLM judges introduces challenges. First, their evaluations can be inconsistent due to prompt sensitivity, inherent biases, or a lack of grounding in objective criteria (Gu et al., 2025). Second, judge models themselves are prone to manipulation; studies show adversarial phrases or stylistic cues (e.g., phrases like "I think") can skew their assessments (Raina et al., 2024; Lee et al., 2024a). Finally, the lack of standardized evaluation protocols and risks like "preference leakage", where judges favour outputs from models they were trained on, underscores the need for multi-judge ensembles, human oversight, and rigorous benchmarking to ensure reliability (Beyer et al., 2025). While promising, LLM-as-a-Judge systems require careful design to balance automation with trustworthiness.

## 5    Metrics and Benchmarks

Modern language models must demonstrate reliability beyond basic performance metrics. As these models are deployed in real-world applications – from healthcare diagnostics to legal document analysis – their behaviour under stress becomes critical. Robustness evaluation answers crucial questions: How does the model fail? When does it fail? Most importantly, *why* does it fail?

The landscape of LLM robustness contains eight interconnected dimensions 2.2, each requiring specialized measurement approaches. Our analysis reveals that the majority of production failures stem from overlooked robustness factors rather than pure accuracy issues. Table 4 distils some essential metrics across 7 categories, providing practitioners with:

- A diagnostic toolkit for model weaknesses

- Prioritization guidance for system improvements

- Benchmarking standards for research comparisons

    **Key Insight:** No single metric tells the whole story. Effective evaluation requires combining complementary measures across multiple dimensions to ensure reliability.

### 5.1 Metrics

Evaluating the multifaceted concept of LLM robustness requires a diverse toolkit of metrics, each designed to capture specific aspects of model behaviour under various challenging conditions. These metrics move beyond simple IID accuracy to quantify performance degradation, generalization ability, consistency, calibration, fairness under duress, and factuality, particularly when models are stressed.

#### 5.1.1 Performance Degradation Metrics

These metrics quantify the drop in performance when an LLM is evaluated on challenging data (adversarial or noisy) compared to its performance on clean, standard data. They provide a direct measure of how much a specific challenge impacts the model's effectiveness.

- Accuracy Drop / F1 Drop / Metric Drop: This metric calculates the difference in model performance between a baseline (clean or standard) dataset and a challenging version of that dataset (e.g., perturbed, adversarial, or noisy) Yuan et al. (2023). This metric directly quantifies how much a given challenge degrades effectiveness when expressed in absolute terms or percentage points. A larger drop indicates lower robustness, while a smaller drop suggests resilience under stress.

- Attack Success Rate (ASR): This metric is central to adversarial robustness evaluation. It measures the percentage of adversarial inputs that successfully cause the intended model failure, such as misclassification in NLU tasks, bypassing safety filters to generate harmful content, or successfully executing a jailbreak prompt (Jung et al., 2025). A higher ASR signifies lower robustness to the specific attack method used. It is a key metric for benchmarks like AdvGLUE++, JailbreakBench, and PromptBench (Zhou et al., 2024a).

- Compliance Ratio / Refusal Rate: These metrics are particularly relevant for evaluating safety and alignment robustness, especially against jailbreaking attempts. It measures how often the LLM generates harmful or forbidden content when prompted with malicious instructions (Varshney et al., 2024; Zeng et al., 2025). Conversely, the Refusal Rate measures how often the model refuses to answer. A high refusal rate for harmful prompts indicates good safety alignment, but a high refusal rate for benign prompts indicates over-defensiveness, a potential negative side-effect of safety training.

#### 5.1.2 Out-of-Distribution (OOD) Performance Metrics

These metrics specifically assess how well an LLM performs when faced with data that differs statistically from its training distribution.

- OOD Accuracy / F1 score: Standard performance metrics like Accuracy or F1-score calculated directly on designated OOD benchmark datasets, such as those included in the BOSS suite (e.g., Dynasent, ANLI, NewsQA) (Yuan et al., 2023). This provides a direct measure of capability on unseen distributions.

Table 4: Different metrics for evaluation of robustness in LLMs.

| Metrics | Specific Category | Dimension Measured | Brief Description |
|---|---|---|---|
| Performance Degradation | Accuracy Drop, F1 Drop (Yuan et al., 2023) | General Robustness | Decrease in standard metrics on challenging vs clean data. |
| | Attack Success Rate (ASR) (Zhou et al., 2024a) | Adversarial Resilience, Safety | Percentage of successful adversarial attacks/jailbreaks |
| | Compliance / Refusal Rate (Varshney et al., 2024; Zeng et al., 2025) | Safety, Alignment | Frequency of generating harmful content or refusing prompts |
| OOD Performance | OOD Accuracy/F1 score (Yuan et al., 2023) | OOD Generalization | Task performance on designated out-of-distribution datasets. |
| | ID-OOD Performance Gap (Yuan et al., 2023) | OOD Generalization | Difference between in-distribution and out-of-distribution performance. |
| | OOD Detection (AU-ROC, Energy Score) (Lu et al., 2024) | OOD Awareness | Ability to distinguish in-distribution versus out-of-distribution inputs. |
| Consistency | Semantic Consistency (Yang et al., 2025b) | Prompt Robustness, Reliability | Stability of output meaning under input paraphrasing or slight modifications. |
| | Consistency Rate (CR) (Nalbandyan et al., 2025) | Prompt Robustness, Sampling Stability | Proportion of identical or equivalent predictions across setups (e.g., prompts, seeds). |
| | Response Consistency (RC) (Rahman et al., 2024) | Prompt Robustness | Frequency of the most common claim across paraphrased prompts. |
| | LLM-based Consistency Score (Saxena et al., 2024) | Sampling Stability, Reliability | LLM judges assess consistency between generated samples. |
| Calibration | Expected Calibration Error (ECE) (Posocco & Bonnefoy, 2021) | Confidence Reliability | Alignment between predicted confidence and actual accuracy |
| | ECE under Distribution Shift (Huang et al., 2024b; LeVine et al., 2024) | OOD Generalization | ECE measured specifically on OOD data |
| Fairness under Stress | Bias Metrics (e.g., FLEX) (Jung et al., 2025) | Fairness Robustness | Evaluation of fairness when models are exposed to bias-inducing prompts. |
| Task-Specific | Code Robustness Checks (Li et al., 2025e) | Code Generation Robustness | Checks for robustness in generated code, like security or correctness features. |
| | Reasoning Accuracy (e.g., ReClor-plus) (Bao et al., 2025) | Reasoning Robustness | Performance on reasoning tasks containing structure variations. |
| Hallucination/Factuality | FactScore, FCH Rate (Rahman et al., 2024) | Factuality, Faithfulness | Verification of generated facts against a trusted knowledge source or context. |
| | FEWL (Wei et al., 2024) | Factuality (Reference-Free) | Weighted agreement among reference LLMs based on expertise. |
| | SelfCheckGPT, MetaQA (Wei et al., 2024) | Factuality/Consistency (Reference-Free) (Chen et al., 2024) | Consistency checks across multiple generated samples or metamorphic prompt variations. |
| | Faithfulness Metrics (NLI/QA-based) | Faithfulness to Source | Consistency between generated text and provided source document. |
| | Hallucination Rate Increase (HRI) | Factuality Robustness under Stress | Change in hallucination rate when exposed to stress conditions. |

- ID-OOD Performance Gap: This metric quantifies the difference in performance between an ID dataset and a corresponding OOD dataset (Yuan et al., 2023). It explicitly measures the drop in generalization capability due to the distribution shift. Studies using benchmarks like BOSS have

shown that this gap can vary significantly and exhibit complex patterns (e.g., linear, piecewise linear, V-shaped) depending on factors like model scale, training steps, and task type, indicating that simply improving ID performance does not always guarantee better OOD performance.

- OOD Detection Metrics (e.g., AUROC, FPR@TPR): These metrics evaluate the model's ability to recognize that an input is OOD, often by thresholding a confidence score derived from model outputs (e.g., token probabilities) or internal representations (Huang et al., 2024a). Common metrics include the Area Under the Receiver Operating Characteristic curve (AUROC) and the False Positive Rate (FPR) at a high True Positive Rate (TPR). While not directly measuring task performance robustness, effective OOD detection is crucial for building reliable systems that can identify and potentially abstain from making predictions on inputs they are likely to handle poorly (Lu et al., 2024). Techniques like Energy Score, originally from classification, have been adapted for this purpose in LLMs and reward models (LeVine et al., 2024).

### 5.1.3 Consistency Metrics

Consistency metrics evaluate the stability of LLM outputs. A consistent model should produce similar outputs for semantically identical inputs and exhibit stable behaviour under non-deterministic sampling. Lack of consistency can signal unreliability, especially for tasks requiring factual accuracy (Nalbandyan et al., 2025).

- Semantic Consistency: Assesses whether the meaning or core information conveyed in the LLM's output remains the same when the input prompt is paraphrased or expressed differently, while preserving the original intent (Nalbandyan et al., 2025). Evaluation can be qualitative or quantitative, often using semantic similarity metrics based on embeddings (e.g., cosine similarity between output embeddings) (Patwardhan et al., 2025).

- Consistency Rate (CR): Proposed within the SCORE framework (Nalbandyan et al., 2025), CR quantifies the agreement among predictions for the same input across different evaluation settings (e.g., 10 different prompts, 5 different random seeds for sampling). It is calculated as the average proportion of agreeing pairs among all possible pairs of predictions for each input instance. For multiple-choice questions (MCQ), agreement means identical predicted letters; for tasks like MATH, it means symbolic equivalence checked via tools like sympy (Nalbandyan et al., 2025).

- Response Consistency (RC): Used in the DefAn benchmark (Bao et al., 2025), this metric specifically measures consistency across multiple (e.g., 15) paraphrased versions of a single prompt. For each set of paraphrased prompts, it calculates the frequency of the most common factual claim generated in the responses. The overall RC is the average of these frequencies.

- Multilingual Consistency: Evaluates whether the model's performance or the semantic meaning of its output is preserved when prompts or inputs are translated into different languages (Qi et al., 2023).

- LLM-based Consistency Score: This approach uses another LLM as a judge (Saxena et al., 2024). It prompts the target LLM multiple times ($n > 1$) for the same input using a non-zero temperature (e.g., 1.0) to induce variability. The judge LLM then compares pairs of generated responses, assessing whether they contradict or support each other. The consistency score is the fraction of pairs deemed consistent.

### 5.1.4 Calibration Metrics

Calibration metrics assess the alignment between an LLM's predicted confidence and its actual likelihood of being correct. Well-calibrated models are crucial for trustworthy decision-making, as their confidence scores reliably indicate the potential accuracy of their outputs. Evaluating calibration under stress, such as distribution shifts, is vital for robustness.

- Expected Calibration Error (ECE): It works by dividing predictions into bins based on their confidence scores (e.g., 10 bins from 0.0-0.1, 0.1-0.2,..., 0.9-1.0). Within each bin, the average confidence and the actual accuracy (fraction of correct predictions) are calculated. ECE is the weighted average of the absolute difference between average confidence and accuracy across all bins, weighted by the number of samples in each bin. A lower ECE indicates better calibration (perfect calibration yields ECE=0). Frameworks like UQ4CT aim to minimize ECE during fine-tuning. (Posocco & Bonnefoy, 2021)

- ECE under Distribution Shift: This involves calculating ECE specifically on OOD datasets or under simulated distribution shifts to assess how calibration holds up under stress. Research suggests that factors like OOD prompts versus OOD responses can impact the calibration of components like reward models differently (LeVine et al., 2024; Huang et al., 2024b).

- Other Metrics include:
  - Brier Score: Measures the mean squared difference between predicted probabilities and actual outcomes (0 or 1). Its other variants include weighted Brier score (Zhu et al., 2024).
  - Area Under the ROC Curve (AUROC): Can be used to assess how well confidence scores discriminate between correct and incorrect predictions when varying a confidence threshold.

### 5.1.5 Fairness Metrics under Stress

These metrics evaluate whether an LLM maintains fairness and avoids amplifying social biases when subjected to adversarial conditions, such as prompts designed to provoke biased responses.

- Concept: Standard fairness evaluation often checks for performance disparities or stereotypical associations on clean data. Fairness robustness evaluation applies similar principles but within a stress-testing context, using benchmarks like FLEX (Jung et al., 2025).

- Specific Metrics: These often involve measuring differences in other robustness metrics (like ASR, refusal rates, toxicity scores, or task accuracy) across different demographic groups when the model is prompted with bias-inducing adversarial inputs (Jung et al., 2025). For example, one might measure if a jailbreak attack is more successful (higher ASR) when the harmful instruction targets a specific demographic group, or if the model refuses benign requests more often for certain groups under adversarial pressure. Metrics derived from causal frameworks, like stratified invariance, can also quantify the model's reliance on protected attributes under intervention (Cotta & Maddison, 2024).

### 5.1.6 Task-Specific Robustness Metrics

Certain applications require robustness dimensions unique to the task domain. Metrics are tailored accordingly.

- Code Generation: Beyond functional correctness, robustness evaluation focuses on aspects like security vulnerabilities or handling of edge cases too (Rahman & Kundu, 2024). Metrics might involve:
  - Static Analysis: Checking for the presence or absence of necessary robustness features like input validation, exception handling, or specific control structures compared to human-written reference code (Li et al., 2025e).
  - Robustness to Prompt Perturbation: Evaluating if code generation remains correct when the natural language instruction contains typos or semantic variations (Gan et al., 2024).
  - Code Hallucination Metrics: Specific metrics from benchmarks like HalluCode, such as Accuracy of Hallucination Existence Recognition (Acc-Rec) (did the model correctly identify if generated code contains hallucination ?) and Accuracy of Hallucination Mitigation (Acc-Mit) (did the model successfully correct the hallucinated code ?) (Liu et al., 2024a).

- Logical Reasoning: Evaluating performance degradation on standard reasoning benchmarks (e.g., ReClor, LogiQA) when the task structure is perturbed, for example, by shuffling the order of multiple-choice options or replacing the correct option with "none of the above" (as done in ReClor-plus, LogiQA-plus) (Bao et al., 2025). Robustness can also be tested against typographical errors in the problem statement using benchmarks like R2ATA (Gan et al., 2024).

### 5.1.7 Hallucination and Factuality Metrics under Stress

This crucial category focuses on measuring an LLM's propensity to generate content that is factually incorrect, inconsistent with provided sources (unfaithful), or nonsensical (hallucination) (Liu et al., 2024a). The evaluation is particularly concerned with how this tendency changes under stressful conditions like OOD inputs, adversarial prompts, or questions probing the model's knowledge boundaries.

- Fact-Checking / Factuality Scores: These metrics assess the alignment of generated content with verifiable world knowledge or a ground truth source.

  - Knowledge Base (KB) / Corpus Comparison: Generated statements are checked against external structured KBs or large factual corpora (Muhlgay et al., 2024). The FACTOR benchmark operationalizes this by creating contrastive (true/false) statements from a corpus and evaluating the LM's likelihood score assigned to the true version versus false alternatives (Muhlgay et al., 2024).
  - Exact Match (EM) (Wang et al., 2024c): Measures the percentage of generated answers that perfectly match a known factual reference answer. It is simple but often too strict, failing to credit semantically correct paraphrases.
  - Atomic Fact Verification (e.g., FactScore, FCH Rate): This approach decomposes a longer generated text into individual factual claims ("atomic facts") and verifies each claim against a reliable source like Wikipedia (Wang et al., 2024c). The final score is often the percentage of verified facts. This allows for fine-grained analysis of long-form generation. The Factual Contradiction Hallucination (FCH) rate used in the DefAn benchmark follows a similar principle (Rahman et al., 2024).
  - FEWL (Factualness Evaluations via Weighting LLMs): A reference-free metric designed for scenarios without gold-standard answers (Wei et al., 2024). It uses multiple external LLMs ("reference LLMs") as proxies. It calculates a truthfulness score by weighting the agreement between the evaluated response and each reference LLM's response based on the reference LLM's estimated "expertise" (disagreement with wrong answers) and "non-laziness" (consistency across similar questions) (Wei et al., 2024).
  - QA/NLI-based Metrics: These leverage auxiliary models. A Question Generation/Answering system can be used to generate questions from the generated text and check if the answers are consistent with the source document. Alternatively, NLI models can classify the relationship (entailment, contradiction, neutral) between generated statements and source text sentences (Chen et al., 2024).

- Faithfulness / Groundedness Scores: These metrics specifically measure whether the generated text accurately reflects and is supported by a given source document. This is critical for tasks like summarization or Retrieval-Augmented Generation (RAG).

  - Methods often overlap with fact-checking (e.g., using NLI or QA models) but focus specifically on adherence to the provided context rather than general world knowledge (Chen et al., 2024). Faithfulness metrics evaluate if all claims in the output can be inferred from the source (Yang et al., 2025a).
  - Mol-Hallu: A domain-specific metric for evaluating faithfulness in molecular comprehension tasks. It uses an entailment model to check if scientific entities mentioned in the generated text are actually supported by the provided molecular descriptions (Li et al., 2025a).

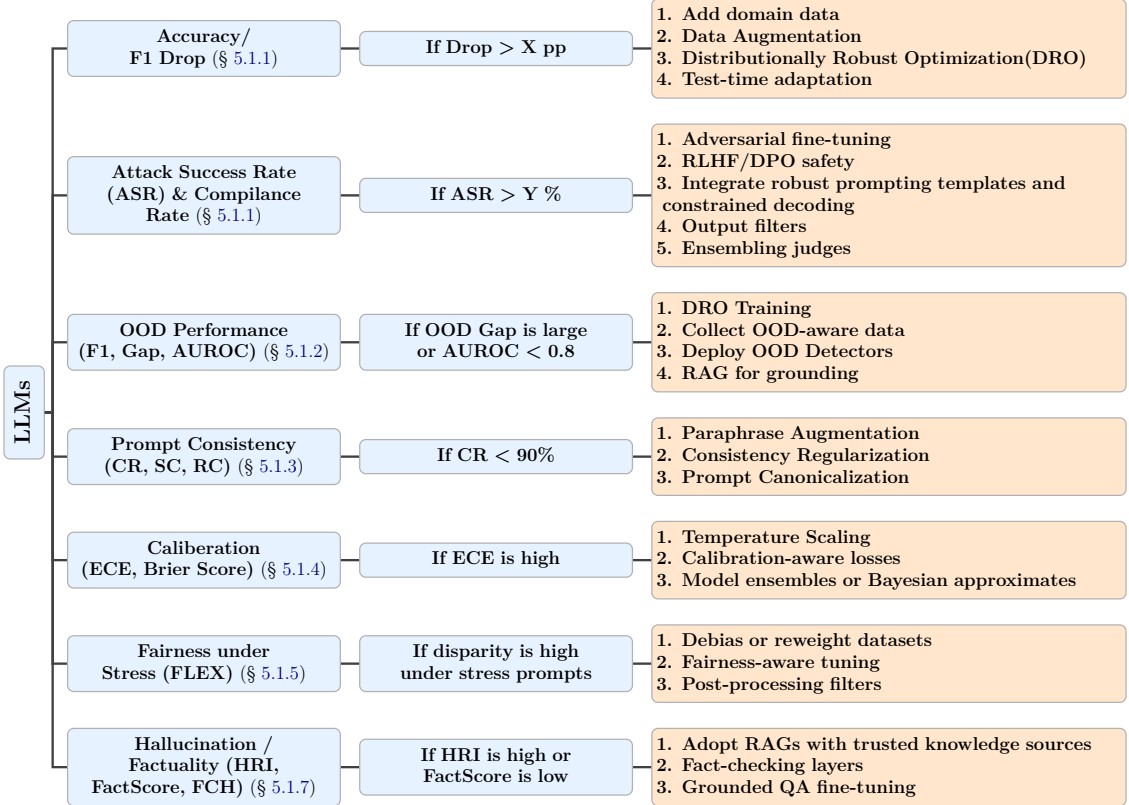

Figure 6: Actionable guidance for LLM practitioners where Performance metrics can be taken from Table 4 and then by identifying the threat model or corresponding weak dimensions, practitioners can apply corresponding strategies from Table 3.

- Consistency-Based Metrics: These metrics leverage the idea that factual information should be stable, while hallucinations might be more variable or contradictory across different generations or prompts.

  - Self-Consistency Checks (e.g., SelfCheckGPT): This approach involves generating multiple responses from the same LLM for a single prompt, typically using temperature sampling (T>0) to introduce diversity (Wei et al., 2024). The consistency (or lack thereof) across these responses is then measured. High variance or contradiction among samples might indicate hallucination, while consistent outputs suggest higher confidence or factuality. Various metrics can be used to quantify this, such as semantic similarity variance or contradiction detection using NLI models (Wei et al., 2024).

  - LLM-based Consistency Score: This method uses a secondary model to evaluate consistency by generating multiple responses (with controlled randomness) to the same prompt, then calculating how often these responses logically align (Saxena et al., 2024). The resulting "consistency score" measures reliability under variable conditions.

  - MetaQA: This method applies metamorphic testing principles (Yang et al., 2025a). It generates variations of the original prompt using synonym or antonym substitutions. It then checks if the LLM's responses to these related prompts exhibit the expected consistency (for synonyms) or inconsistency (for antonyms). Deviations from expected patterns can signal factual errors or hallucinations (Yang et al., 2025a).

- Uncertainty as Indicator: The model's own uncertainty estimates can sometimes serve as a proxy for hallucination risk.

– Confidence Scores: Using metrics derived from the model's output probabilities, such as token-level probabilities, sequence log-probability, or entropy (Mahaut et al., 2024). The assumption is that lower confidence might correlate with a higher likelihood of hallucination. However, this requires the model to be well-calibrated for the confidence scores to be reliable indicators.

– Energy Score: While primarily used for OOD detection, the energy score calculated for reward models can indicate when the model is operating on inputs (prompts or responses) far from its training distribution, suggesting its reward estimate (which might relate to factuality) could be unreliable (LeVine et al., 2024).

- Evaluating Increase in Hallucination under Stress: A key aspect of robustness is understanding how hallucination behaviour changes under pressure. This requires comparing hallucination rates measured using the metrics above under standard conditions versus stress conditions.

  – Methodology: Apply a chosen hallucination metric (e.g., FCH rate, FEWL score, SelfCheckGPT score) to a standard benchmark (e.g., TruthfulQA) and then again to a stressed version of that benchmark (e.g., TruthfulQA prompts with added noise, OOD context, or adversarial perturbations).

  – Metric: The Hallucination Rate Increase (HRI) can be defined as HRI = Hallucination Rate (Stress) - Hallucination Rate (Standard). A significantly positive HRI indicates a failure in factuality robustness, meaning the model becomes more prone to hallucination when challenged. This requires benchmarks specifically designed or adapted for stress testing factuality (discussed in Section 2.2).

## 5.2 Benchmarks

Complementing the metrics described above, a growing number of benchmarks and evaluation frameworks are being developed to systematically assess LLM robustness across its various dimensions. Table 5 provides an overview of the benchmarks discussed. These benchmarks provide standardized datasets and protocols for testing models under specific types of stress.

### 5.2.1 Adversarial Benchmarks

These benchmarks are designed to evaluate an LLM's resilience to intentionally crafted inputs aimed at causing model failure, such as misclassification, generation of harmful content, or bypassing safety mechanisms.

- AdvGLUE / AdvGLUE++: These benchmarks are adversarial versions of the popular GLUE (General Language Understanding Evaluation) benchmark. They apply various textual adversarial attack methods (e.g., word substitutions using synonyms, character-level perturbations, sentence paraphrasing) to the original GLUE tasks (like sentiment analysis, NLI). AdvGLUE++ (Wang et al., 2021) expands upon AdvGLUE with more adversarial examples, targeting newer LLMs. They serve as a standard for evaluating the robustness of fundamental language understanding capabilities against common textual attacks.

- PromptBench: Unlike benchmarks focusing on perturbing input samples, PromptBench (Zhang, 2023) specifically evaluates the robustness of LLMs to adversarial prompts (the instructions given to the model). It includes a framework for dynamically generating adversarial prompts using character, word, sentence, and semantic-level attacks and applies these perturbed prompts across various tasks (sentiment analysis, NLI, QA, translation, math) and datasets. This targets the sensitivity of LLMs to the way instructions are phrased.

- Jailbreak Benchmarks (e.g., JailbreakBench, AdvBench subset): These benchmarks consist of prompts specifically designed to circumvent an LLM's safety alignment training and elicit responses that are harmful, unethical, illegal, or otherwise prohibited by the model's safety guidelines (Xhonneux et al., 2024). Examples include prompts using role-playing scenarios, hypothetical situations, or character encodings to trick the model. Evaluating performance on these benchmarks (often using ASR or Compliance Ratio) is crucial for assessing the effectiveness of safety measures.

- R2ATA (Gan et al., 2024): This benchmark focuses on the robustness of LLM reasoning capabilities. It uses the Adversarial Typo Attack (ATA) algorithm to introduce minimal, targeted typographical errors into reasoning problems (from datasets like GSM8K, BBH, MMLU). It then measures the degradation in the LLM's ability to perform correct step-by-step reasoning (Chain-of-Thought) and arrive at the right answer.

Adversarial benchmarks, while primarily focused on inducing misclassification or safety failures, can inadvertently trigger hallucinations. Attacks that manipulate reasoning processes or push models into unusual states via jailbreaking prompts (Gan et al., 2024; Zhou et al., 2024a) might cause them to generate factually incorrect or nonsensical outputs as a byproduct of the attack's success or the model's attempt to handle the confusing input. Evaluating factuality metrics on adversarial datasets can provide insights into how robust a model's factual grounding is when under direct attack.

### 5.2.2 OOD Benchmarks

OOD benchmarks are designed to evaluate how well LLMs generalize their capabilities to data drawn from distributions that differ from their training data. This tests robustness against natural shifts encountered in the real world.

- BOSS (Benchmark suite for Out-of-distribution robustneSS): BOSS is a notable effort to create a more challenging and holistic OOD evaluation suite for NLP. It covers five diverse tasks: Sentiment Analysis (SA), Toxic Detection (TD), NLI, Named Entity Recognition (NER), and Extractive Question Answering (EQA) (Yuan et al., 2023). For each task, BOSS includes one large, diverse In-Distribution (ID) dataset (e.g., Amazon reviews for SA, MNLI for NLI, SQuAD for EQA) and three corresponding OOD datasets (Yuan et al., 2023). The OOD datasets are carefully selected based on a protocol that prioritizes: (1) distinct data sources/domains to ensure low semantic similarity (measured via SimCSE) with the ID set and (2) evidence of a significant performance drop for a baseline model trained on the ID set, indicating a challenging distribution shift (Yuan et al., 2023). Examples of OOD datasets included are Dynasent, SemEval (SA); ANLI, ContractNLI, WANLI (NLI); AdvQA, NewsQA, SearchQA (EQA) (Yuan et al., 2023).

- ANLI (Adversarial NLI): Created through an iterative human-and-model-in-the-loop process, ANLI contains NLI examples specifically designed to be difficult for contemporary models. Due to its challenging nature and distinct creation process, it is frequently used as an OOD benchmark for NLI task generalization (Yuan et al., 2023).

- Dynasent: A sentiment analysis dataset generated dynamically using adversarial methods to create challenging examples that models trained on standard sentiment datasets might misclassify (Yuan et al., 2023).

- WANLI: An NLI dataset synthesized using GPT-3, focusing on challenging linguistic patterns that models often struggle with in datasets like MNLI (Yuan et al., 2023).

- OODRobustBench (Li et al., 2024a): This is a comprehensive benchmark to evaluate adversarial robustness under dataset and threat distribution shifts. Further, this benchmark demonstrates that adversarial robustness suffers significant degradation under distribution shifts, yet in-distribution robustness correlates strongly and linearly with out-of-distribution robustness.

### 5.2.3 Consistency/Prompt Robustness Benchmarks

These benchmarks focus on evaluating the stability of LLM outputs when faced with variations in prompt formulation or non-deterministic sampling, testing whether the model's behaviour is sensitive to superficial changes.

- SCORE (Systematic Consistency and Robustness Evaluation): SCORE (Nalbandyan et al., 2025) provides a framework for non-adversarial robustness evaluation. It assesses models on existing benchmarks (like MMLU-Pro, AGIEval, MATH) under three specific perturbation scenarios :

Table 5: Different Benchmarks for evaluation of robustness in LLMs.

| Target Dimension | Benchmarks | Example Dataset / Key Component | Evaluation Approach |
|---|---|---|---|
| Adversarial Resilience | AdvGLUE / AdvGLUE++ | Modified SST-2, MNLI, QQP, etc. | Perturbed versions of GLUE using textual attacks |
| Adversarial Resilience (Prompts) | PromptBench | Uses prompts with datasets like GLUE, SQuAD, GSM8K, GSM-PLUS | Dynamic adversarial attacks on prompts (char, word, sentence, semantic) |
| Adversarial Resilience (Safety) | JailbreakBench, AdvBench | Custom jailbreak prompts | Prompts designed to bypass safety filters |
| Adversarial Resilience (Reasoning), Noise | R2ATA | Perturbed GSM8K, BBH, MMLU | Adversarial typographical errors in reasoning problems |
| OOD Generalization | BOSS | IID: Amazon, SQuAD, MNLI, CC, FewNerd; OOD: Dynasent, ANLI, NewsQA, etc. | Curated ID/OOD datasets based on semantic dissimilarity & performance drop protocol. |
|  | OODRobustBench | OODRobustBench | Predicts the maximum OOD robustness achievable by extrapolating existing $l_p$-based robust training techniques to hypothetical models with perfect ID robustness. |
|  | WANLI | ANLI dataset | Adversarially collected NLI examples. |
| Consistency, Prompt Robustness | SCORE | MMLU-Pro, AGIEval, MATH | Evaluates consistency across prompt paraphrasing, sampling seeds, choice order permutations. |
| Fairness under Stress, Adversarial Resilience | FLEX | Modified BBQ, CrowS-Pairs, StereoSet | Adversarial prompts (persona, competing objectives, text attacks) on fairness datasets |
| Task-Specific (Reasoning) | ReClor-plus, etc. | Modified ReClor, LogiQA | Structural variations (shuffled options, or adding "none of the above") on reasoning datasets. |
| Task-Specific (Code) | CoderEval | CoderEval benchmark | Used to analyze code robustness aspects like missing checks vs. human code |
| Factuality/Hallucination | TruthfulQA | TruthfulQA dataset | Questions targeting common misconceptions to test truthfulness vs. imitation |
|  | FELM | FELM dataset | Fine-grained, segment-level factuality annotations with error types & references |
|  | FACTOR | Wiki-FACTOR, News-FACTOR, etc. | LM likelihood evaluation on contrastive (true vs. false) statements derived from a corpus |
|  | DefAn | DefAn dataset | Large dataset evaluating FCH, PMH, and RC across paraphrased prompts |
|  | UMWP | UMWP dataset | Tests if LLMs identify unanswerable problems or hallucinate solutions |
| (Recognition) | HaluEval | HaluEval dataset (derived from HotpotQA, CNN/DM, Alpaca, etc) | Large dataset of hallucinated/correct samples; tests LLM ability to classify hallucination |
| (Code) | HalluCode / CodeHalu | HalluCode, Code-HaluEval datasets | Benchmarks with taxonomies for evaluating code-specific hallucinations |
| (Multimodal) | AMBER | AMBER dataset (custom annotated images) | LLM-free evaluation of existence, attribute, relation hallucinations in MLLMs |
| (Multimodal) | HQHBench / ODE | HQHBench dataset; ODE framework | Meta-evaluation of benchmark quality (HQHBench); Dynamic open-set generation (ODE) |

- Prompt Robustness: Testing with 10 different non-adversarial, semantically equivalent prompts for each question, varying structure and including Chain-of-Thought (CoT) variations.

- Non-Greedy Inference: Evaluating consistency across 5 runs using the same prompt but with temperature sampling (T=0.7) and different random seeds.

- Choice Order Robustness: For MCQ datasets, shuffling the order of answer choices while keeping the correct answer logically in the same position. SCORE reports the range (min/max) of accuracy across these scenarios and the overall Consistency Rate (CR) (Nalbandyan et al., 2025).

- Paraphrasing Benchmarks: Methodologies that involve generating multiple paraphrased versions of instructions or queries and measuring the consistency of the LLM's responses (Barbero et al., 2025). This directly tests robustness to linguistic variation.

- Structural Variation Tests: Evaluating how changes in prompt structure, such as altering the position of the question within the prompt (beginning, middle, end), affect model performance and consistency (Nalbandyan et al., 2025).

### 5.2.4 Fairness/Bias Robustness Benchmarks

These benchmarks are specifically designed to assess whether LLMs maintain fairness and avoid biased outputs when subjected to stress or adversarial prompts intended to provoke bias.

- FLEX (Fairness Benchmark in LLM under Extreme Scenarios): FLEX (Jung et al., 2025) focuses on the robustness of fairness. It takes samples from existing fairness benchmarks (like BBQ, CrowS-Pairs, StereoSet) where baseline models typically provide unbiased responses. It then applies adversarial prompts using techniques like persona injection (instructing the LLM to adopt a biased persona), competing objectives (presenting conflicting goals that might induce bias), or text attacks (subtle manipulations of the input) (Jung et al., 2025). FLEX evaluates whether the model maintains its unbiased stance even under these extreme, bias-inducing conditions.

### 5.2.5 Task-Specific Robustness Benchmarks

These benchmarks evaluate robustness in the context of particular downstream applications where unique failure modes might exist.

- Reasoning:
  - ReClor-plus, LogiQA-plus, LogiQAv2-plus (Bao et al., 2025): These extend standard logical reasoning datasets (ReClor, LogiQA, LogiQAv2) by introducing structural variations to the multiple-choice questions. These variations include randomly shuffling the answer options or replacing the correct answer with a "none of the other options is correct" choice, testing if the model relies on superficial patterns or truly understands the logic (Bao et al., 2025).
  - R2ATA (Gan et al., 2024): Assesses reasoning robustness specifically against adversarial typographical errors introduced into the problem statements of reasoning tasks (Gan et al., 2024).

- Code Generation:
  - CoderEval (Li et al., 2025e): While primarily a code generation benchmark, it has been used in studies focusing on code robustness issues beyond simple functional correctness, such as identifying missing input validation or error handling checks by comparing generated code to human references (Li et al., 2025e).
  - Benchmarks evaluating robustness to perturbations in the natural language instructions provided for code generation tasks are also emerging (Barbero et al., 2025).

### 5.2.6 Hallucination/Factuality Benchmarks

This category includes benchmarks explicitly created to measure an LLM's tendency to hallucinate or its ability to adhere to factual knowledge.

- TruthfulQA: This benchmark (Cecchini et al., 2024) is designed to measure whether LLMs are truthful in generating answers to questions where humans often hold common misconceptions. It tests if models avoid repeating false beliefs prevalent in their training data, distinguishing truthfulness from mere imitation (Yang et al., 2025a). It is also used widely for evaluating factuality robustness with an updated version, TruthfulQA-Enhanced (Yang et al., 2025a).

- HaluEval: A large-scale benchmark (35,000 samples) focusing on the LLM's ability to recognize hallucinations (Li et al., 2023). It includes generated and human-annotated samples across QA, knowledge-grounded dialogue, text summarization, and general user queries. Models are evaluated on their ability to correctly classify provided text as containing hallucinations ("Yes") or not ("No")

- FELM (Factuality Evaluation of LLMs): FELM (Chen et al., 2023b) provides fine-grained, segment-level factuality annotations for LLM-generated responses. For each segment, human annotators provide a factuality label (correct/incorrect), identify the error type if incorrect, and list reference links supporting the judgment. It covers diverse domains including world knowledge, science, math, and reasoning.

- FACTOR: This benchmark (Muhlgay et al., 2024) automatically transforms a given factual corpus (e.g., Wikipedia, news articles) into a set of evaluation instances. Each instance consists of a context, a factually correct completion, and several plausible but factually incorrect variations (generated based on different error types like entity substitution, negation, and numerical errors). It evaluates an LM by measuring whether it assigns a higher likelihood (probability) to the factually correct completion compared to all the incorrect alternatives in a generation setting (Muhlgay et al., 2024).

- DefAn: A large dataset ( 75k prompts) across eight domains (including Sports, Census data, Nobel Prize) designed to elicit definitive, concise factual answers (Rahman et al., 2024). It evaluates three aspects: Factual Contradiction Hallucination (FCH rate - % of responses with incorrect facts), Prompt Misalignment Hallucination (PMH rate - % of responses deviating from prompt instructions/format), and Response Consistency (RC - consistency of claims across 15 paraphrased prompts) (Rahman et al., 2024).

- UMWP (Unanswerable Math Word Problem): This benchmark (Sun et al., 2024a) evaluates hallucination in the context of mathematical reasoning. It consists of math word problems that are intentionally designed to be unanswerable (e.g., due to missing information). It tests whether LLMs correctly identify these problems as unanswerable or if they hallucinate by attempting to provide a numerical solution.

- Code Hallucination Benchmarks (HalluCode, CodeHalu, Collu-Bench): These benchmarks (Liu et al., 2024a) are specifically designed for the code domain. They often include taxonomies of code-specific hallucinations (e.g., intent conflicting, context inconsistency, dead code, knowledge conflicting (Liu et al., 2024a); mapping, naming, resource, logic hallucinations (Tian et al., 2025)). They evaluate an LLM's ability to generate correct code and/or recognize hallucinations in generated code snippets. Collu-Bench, for instance, focuses on predicting hallucination locations based on generation log probabilities and execution feedback (Jiang et al., 2024).

- Multimodal Hallucination Benchmarks (AMBER, POPE, MME, HQHBench, ODE, etc.): These benchmarks evaluate hallucinations in MLLMs, where the generated text might conflict with the provided visual (or other modality) input (Bai et al., 2025b).

  - AMBER (Bai et al., 2025b; Wang et al., 2024b) is notable for being an LLM-free benchmark. It uses carefully annotated images and an automated pipeline to evaluate MLLM hallucinations across both generative and discriminative tasks, covering existence (object presence), attribute (object properties), and relation (spatial/contact relations between objects) hallucinations without relying on another LLM for judgment (Wang et al., 2024b). While other benchmarks like ODE (Tu et al., 2024) propose a dynamic, open-set protocol to generate novel evaluation samples and mitigate data contamination.

- Evaluating Hallucination under Stress: While the benchmarks listed above are primarily designed to elicit or detect hallucinations, they form the basis for evaluating factuality robustness under stress. This can be achieved by:

  - Applying stress conditions (e.g., adding noise, using OOD contexts, applying adversarial perturbations) to the prompts or inputs of these benchmarks (e.g., stressing TruthfulQA or HaluEval inputs).

- Measuring the change in hallucination rates using appropriate metrics ( under standard conditions versus stress conditions). Benchmarks like UMWP or DefAn inherently contain challenging conditions designed to probe these factuality limits.

The diversity of these hallucination benchmarks highlights the complexity of the phenomenon. Some test the generation of truthful content (TruthfulQA, FACTOR, DefAn), while others test the recognition of flawed content (HaluEval, HalluCode recognition tasks). They cover different domains (general knowledge, dialogue, code, math, multimodal) and rely on different evaluation paradigms (reference-based, reference-free, likelihood-based, classification-based). A comprehensive assessment of an LLM's factuality and hallucination tendencies requires leveraging multiple benchmarks that cover these different facets (Liu et al., 2024a). Furthermore, the field is actively evolving to address in traditional static benchmarks. There is a noticeable trend towards developing benchmarks that are dynamic (generating test cases on-the-fly to prevent contamination, e.g., ODE, DyVal) (Tu et al., 2024), LLM-free (avoiding reliance on potentially unreliable LLM judges, e.g., AMBER) (Wang et al., 2024b). These advancements aim to improve the reliability and validity of LLM robustness evaluations.

# 6 Challenges and Future Work

The field has made notable progress in identifying and characterizing robustness challenges in LLMs. While awareness of these vulnerabilities is now widespread, and numerous mitigation strategies and evaluation frameworks have emerged, even state-of-the-art LLMs demonstrate brittleness across critical dimensions. Robustness does not consistently correlate with model scale as larger models are not inherently more robust, and scaling trends often show weak or inconsistent patterns (Zhou et al., 2024d; Yang et al., 2024d). Safety-aligned models remain susceptible to sophisticated jailbreaking attempts, while performance may deteriorate sharply with minor distribution shifts or input perturbations (Qi et al., 2025; Oh & Demberg, 2025; Yuan et al., 2023). Ultimately, achieving comprehensive and reliable robustness persists as a fundamental challenge for the field.

## 6.1 Major Challenges and Limitations

Several key challenges that hinder progress towards truly robust LLMs are:

1. Scalability of Defences: Many promising defence mechanisms, particularly those involving adversarial example generation during training (AT) or complex inference-time checks, are computationally expensive and may not scale efficiently to trillion-parameter models (Lee et al., 2024b; Howe et al., 2025). Ensuring robustness improvements keep pace with model scaling is crucial but not guaranteed.

2. Robustness-Utility Trade-offs: Aggressively optimizing for robustness can often negatively impact the model's general capabilities or performance on clean, standard tasks (Lin et al., 2024b; Lee et al., 2024b). The alignment tax associated with RLHF is a prime example of it. Finding the right balance and developing methods that improve robustness without sacrificing utility is a critical ongoing challenge.

3. Evaluation Gaps: Current evaluation methodologies have limitations. There is a need for more comprehensive benchmarks covering a wider range of robustness dimensions simultaneously, including more subtle and complex failure modes (e.g., sophisticated shortcuts). Evaluating the robustness of generative capabilities remains particularly difficult compared to classification tasks (Ailem et al., 2024). Ensuring benchmarks are realistic, efficient, and resistant to "teaching to the test" is also vital.

4. Theoretical Understanding: Our theoretical understanding of why LLMs exhibit certain robustness failures like shortcut learning, understanding how model architecture (e.g., attention mechanisms, MoE) influences robustness and vulnerabilities or the alignment tax and why certain defences work is still developing (Ye et al., 2024). Deeper theoretical insights could guide the development of more principled and effective mitigation strategies.

5. Real-world Deployment: Translating robustness improvements observed on static benchmarks into reliable performance in dynamic, open-world environments with continuous distribution shifts and novel, unforeseen threats remains a significant hurdle. Bridging this gap is essential for trustworthy deployment.

6. Robust Evaluation Methodologies: A critical need exists for more reliable, comprehensive, and standardized benchmarks and evaluation protocols. This includes developing robust LLM judges, methods for verifying judge outputs, and metrics that capture real-world failure modes more effectively.

## 6.2 Future Research Directions

Addressing these challenges points towards several promising avenues for future research:

1. Causal and Invariant Learning: Moving beyond correlational learning towards models that understand underlying causal relationships or learn representations that are inherently invariant to changes in domain or distribution (Ye et al., 2024). This could lead to more fundamentally robust models less reliant on spurious cues.

2. Compositional Robustness: Investigating how models behave under combinations of different perturbations (e.g., noisy input + OOD context + adversarial prompt) and developing methods to ensure robustness in such complex scenarios.

3. Hybrid Approach: Exploring hybrid approaches that combine multiple processing steps, as well as investigating methods to optimize their computational efficiency, are promising directions for advancing the field and ensuring the safe and reliable deployment of LLMs in a wide range of real-world applications.

4. Efficient Robust Training and Alignment: Developing more scalable adversarial training methods (e.g., improving continuous AT), more efficient robust optimization techniques, and alignment methods (like RLHF variants) that explicitly minimize the alignment tax or are inherently more robust to reward model imperfections (e.g., reward-robust RLHF) (Yan et al., 2024b).

5. Adaptive and Automated Evaluation: The field must prioritize evaluating evaluation methods themselves, specifically, by assessing the reliability, validity, and biases inherent in existing benchmarks and automated metrics. Furthermore, there is a critical need to develop dynamic evaluation platforms and automated red-teaming methods capable of continuously probing models for weaknesses, adapting to emerging attack strategies, and delivering realistic, ongoing assessments of robustness.

6. Improving Benchmarks: To advance benchmarking practices, future efforts should prioritize comprehensive benchmarks that cover a broader range of tasks, domains, languages, modalities (e.g., text, images, or audio), and robustness challenges. These benchmarks should dynamically generate test cases during evaluation to prevent data leakage and rigorously assess adaptability. Equally critical is ensuring benchmarks mirror real-world conditions, such as natural distribution shifts, complex user interactions, and multifaceted failure modes, rather than relying on simplistic, artificial perturbations. Finally, rigorous validation is needed to confirm that benchmarks accurately measure the capabilities they claim to evaluate, avoiding misaligned or inflated performance metrics.

7. Robustness in Multimodal and Agentic Systems: Extending robustness research to the unique challenges of MLLMs and emerging LLM-based agent systems, considering factors like tool use, memory, and interaction with environments (Jiang et al., 2025; Yu et al., 2025).

8. Interpretability for Robustness: Leveraging interpretability techniques to better understand why models fail in specific robustness scenarios (e.g., identifying responsible components (Yang et al., 2025b) or internal representations of truth/lies (Bürger et al., 2024)) and using these insights to design targeted interventions or more robust architectures like a reconstruction attack (Stacey et al., 2024; 2022).

In conclusion, while LLMs have demonstrated transformative capabilities, ensuring their robustness remains critical for safe and responsible real-world adoption. Though the field has advanced significantly in diagnosing vulnerabilities and proposing mitigation strategies, fundamental challenges persist. Future progress will depend on four key priorities: (1) deepening theoretical insights into causality and invariance, (2) developing efficient and robust training frameworks that scale with model size, (3) creating adaptive evaluation systems that address evolving failure modes, and (4) extending robustness principles to emerging paradigms like MLLMs and autonomous AI agents. Addressing these challenges systematically will be essential to transition from promising prototypes to trustworthy, reliably deployable AI systems.

## 7 Conclusion

The field of robust LLMs has moved rapidly from adapting existing ML techniques to developing LLM-specific strategies that address unique challenges like instruction following, alignment, and scalability. In light of these developments, this survey offers a comprehensive overview of ~~Large Language Model (LLM)~~ robustness, a critical factor in ensuring their trustworthiness and reliable deployment. We define LLM robustness as a multi-faceted concept that encompasses resilience to adversarial attacks, generalization to out-of-distribution data, stability under prompt variations, tolerance to noisy inputs, and output consistency. Our analysis highlights that non-robustness arises from systemic issues spanning the entire LLM lifecycle. We then categorize methods to mitigate these vulnerabilities into four groups based on their deployment stage. Furthermore, we evaluate widely used metrics and benchmarks that have been employed to assess LLM performance across diverse dimensions in recent research. Finally, we outline key challenges for future work, with the goal of advancing LLM reliability and enabling their safe, real-world application.

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
