# OpenReview forum: "Robustness in Large Language Models: A Survey of Mitigation Strategies and Evaluation Metrics"
_TMLR — Accepted by TMLR_

### Review · Reviewer_eekr · 2025-07-16

**Summary Of Contributions:**

This survey paper provides a comprehensive overview of robustness challenges in llms. The authors define llm robustness as a multi-faceted concept encompassing resilience to adversarial attacks, out-of-distribution generalisation, prompt stability, noise tolerance and output consistency. They then organise current literature into a coherent framework spanning the entire llm lifecycle.

**Audience:**

Yes

**Broader Impact Concerns:**

None identified.

**Claims And Evidence:**

Yes

**Requested Changes:**

1. Include a detailed search protocol (databases, date range, search strings), explicit inclusion and exclusion criteria or a PRISMA-style flow diagram
2. Provide crisper definition for adversarial attacks, jailbreaks and prompt injections
3. Include simple quantitative cost–benefit analyses for key threat model types
4. Add actionable guidance for practitioners (I can suggest either a decision matrix or flowchart, but even simple key takeaway sentences would be valuable)

**Strengths And Weaknesses:**

## Strengths
1. Provides a clear working definition of llm robustness that synthesizes multiple perspectives from the literature, addressing a gap in consistent terminology across the field
2. The categorisation is structured and the classification is a path forward to more standardised and consistent terminology in the field
3. The lifecycle approach is very practical and not something I've seen in many survey papers

## Weaknesses
1. The search methodology seems to be absent. No search protocol, inclusion/exclusion criteria or PRISMA-style flowchart. Without these the reproducibility is limited and it's hard to gauge whether there are gaps in the survey approach.
2. Some of the writing could be sharpened and more coherent, for example: "Large Language Models (LLMs)" is defined four times across the paper, NLI is defined three times. It only needs to be defined once.
3. The definitions and delinations of adversarial attacks vs jailbreaks vs prompt injections are not clear to me, yet a lot of the phrasing seems to suggest these are being treated as distinct. Although the authors do mention that there can be overlaps (and I agree that they are almost impossible to avoid), it would be good to get some clarification in this instance.
4. The authors do go into trade-offs between different approaches, but there's no deep systematic critical evaluation of relative merits. I would have been interested in head-to-head comparisons  or even fairly simple scalability or cost-benefit quantifications. This would help both researchers and practitioners answers questions like "which defense mechanisms are most effective for specific threat models" and result in more informed decisions and increase the value of the paper
5. There is also a lack of more direct guidance for practitioners. Table 2 helps with this to some extent but it's not explicit in their recommendation. This ties back to my previous point about cost-benefit analyses to make the findings more actionable to practitioners.

---

> ### Author Response · Authors · 2025-09-30
> **Response to Reviewer eekr**
>
> Comment:
>
> Thank you for your thoughtful review and constructive feedback. We appreciate your positive comments on the paper's structure and readability, and we have made revisions to strengthen the manuscript. A revised version of the PDF, incorporating all updates, has also been uploaded for your reference. We have provided changes corresponding to each of your suggestions as follows:
>
> > Weakness 2: Some of the writing could be sharpened …
>
> In addition to addressing the specific points raised, we have reviewed the entire manuscript to ensure sharper and more coherent writing. Redundant definitions and repeated explanations have been removed, and terms such as “Large Language Models (LLMs)” and “NLI” are now defined only once at their first mention, with consistent usage thereafter. These refinements improve readability and avoid unnecessary repetition across the paper.
>
>
> Other weaknesses are resolved in the requested changes.
>
> > Requested changes 1: Include a detailed search protocol (databases, date range, search strings), explicit inclusion and exclusion criteria or a PRISMA-style flow diagram
>
> We have addressed your feedback on the survey methodology by adding $\textbf{Section 1.1}$ in the paper, with the changes highlighted in violet. The revised content, placed after the Introduction, now includes a table summarising the databases and sources consulted. We have also added a PRISMA-style flow diagram with explicit inclusion and exclusion criteria, as you advised.
>
> > Requested changes 2: Provide a crisper definition for adversarial attacks, jailbreaks and prompt injections
>
> We have revised $\textbf{Section 2}$ in the updated version of the paper to provide crisper definitions of adversarial attacks, prompt injections, and jailbreaks, as you advised. The changes are highlighted in violet. For ease of reference, the definitions for adversarial attacks and prompt injections are on $\textb{page 5}$, and the definition of jailbreaks is in $\textbf{Section 2.2.1, first paragraph.}$
>
> > Requested changes 3: Include simple quantitative cost–benefit analyses for key threat model types
>
> We have addressed your feedback on trade-offs between different approaches by incorporating a quantitative cost-benefit analysis for key threat model types. This is presented in $\textbf{Table 3}$ at the end of $\textbf{Section 4}$, where the key threats are also related to the primary dimensions of robustness discussed in the paper.
>
> > Requested changes 4: Add actionable guidance for practitioners (I can suggest either a decision matrix or flowchart, but even simple key takeaway sentences would be valuable).
>
> We are grateful for your insightful suggestion to include actionable guidance for practitioners. In response, we have added $\textbf{Figure 6}$ at the end of $\textbf{Section 5}$, which presents a flowchart outlining the steps for evaluating the model across different metrics to identify the applicable threat model. Based on this analysis, and with reference to $\textbf{Table 3}$, the flowchart further recommends appropriate mitigation strategies.

---

### Review · Reviewer_LBK7 · 2025-08-03

**Summary Of Contributions:**

This survey paper examines the robustness of large language models (LLMs). Previous research has explored this topic from multiple angles, including adversarial attack where prompts are intentionally crafted to elicit harmful or incorrect responses, as well as issues like hallucinations and domain shifts. The survey begins by providing a comprehensive definition of LLM robustness. It then analyses the different factors contributing to non-robust behaviour and reviews strategies to address these weaknesses. Additionally, the paper discusses existing benchmarks and metrics used to evaluate the reliability of LLMs.

**Audience:**

Yes

**Claims And Evidence:**

Yes

**Requested Changes:**

1. Please propose a composite benchmark (like a leaderboard) that can be used to evaluate the robustness of open-source LLMs across the dimensions mentioned in the survey. Evaluate at least a few of the popular opensource models (at least the smaller ones) on this new benchmark.

2. Please fix the issues (1,2,3) mentioned in weaknesses above.

3. Please fix the typos mentioned in the previous section.

**Strengths And Weaknesses:**

Strengths:

1.	The survey is well-motivated. Structuring the discussion of LLM robustness around definition, causes, evaluation methods and mitigation strategies provides a foundation for advancing the field and developing robust LLMs.

Weaknesses:

1.	The paper offers valuable insights about various factors that affect the robustness of LLMs. However, adding references wherever possible would strengthen these discussions and make them more concrete. For example, on page 8, when discussing biases (e.g., lexical, positional), citing relevant papers that demonstrate these biases in practice would improve credibility.
2.	Section 3 describes the sources that contribute to reduced robustness in LLMs. Thus, the appropriate title should be something like “Sources of Robustness Failures in LLMs”.
3.	The paper discusses “Accuracy Drop” in section 5.1.1 and the “IID-OOD Performance Gap” in section 5.1.2, but their definitions seem very similar. Please clarify the distinction between these two metrics, as they currently appear redundant.
4.	The key contributions of this survey lies in its effort to define the scope of LLM robustness and unify diverse existing works on its various subaspects under a single framework. Additionally, the paper effectively categorizes the literature by highlighting different sources of non-robustness, mitigation strategies, and evaluation benchmarks across multiple dimensions.
To enhance the paper’s contributions for the research community, it would be beneficial to propose a composite benchmark (built by aggregating multiple existing benchmarks) that can holistically evaluate all dimensions of robustness discussed in the paper. Furthermore, evaluating popular open-source models such as LLaMA, Mistral, and Qwen, etc, on this proposed benchmark would offer concrete insights and enhance the practical relevance of the survey.

Suggestions and Typos:

1. I understand the motivation behind including Figure 2 and Figure 3; however, on their own, these figures do not clearly convey the intended points. One still needs to refer to the text description to understand their purpose. For example, in Figure 2, the tree illustration uses roots to represent various dimensions of robustness, but it is unclear what the tree itself signifies. If the tree is meant to represent the LLM, then it would be helpful to explicitly label the tree trunk as “LLM” to clarify its meaning.
Similarly, to enhance the clarity in Figure 3, authors may onsider adding a descriptive title like “Synergies and trade-offs in building robust LLMs”. Incorporating labeled arrows like “Fairness vs Generalization” between competing objectives would help better communicate the underlying trade-offs and make the figure more self-explanatory.
2. Figure 1 - Future Work says "Will be updated later", which appears to be a placeholder. Please update this appropriately.
3.	Page 7. The heading Inference Related does not have any accompanying description or content.
4.	page 9 – Phrasing issue:
“The process of training and aligning LLMs can introduces such vulnerabilities:” should be revised.
5.	Section 3.2.2 – 2nd line – the work robustness sources seems weird:
“the process itself can introduce robustness Sources in the following ways”. Pease rephrase for clarity.
6.	Page 31: Robust Evaluation Strategies -> the following passage has  unexplained numbers (I believe authors wanted to cite these subsections) like 52, 59 etc. in the text. Please fix this.

---

> ### Author Response · Authors · 2025-09-30
> **Response to Reviewer LBK7- part 1**
>
> Comment:
> We sincerely appreciate your detailed review and recognition of our work's merits. A revised version of the PDF, incorporating all updates with changes highlighted in violet, has been uploaded. Please find our responses as follows:
>
> > Weakness 1: The paper offers valuable insights about various factors that affect the robustness of LLMs. However, adding references wherever possible would strengthen these discussions and make them more concrete.
>
> Thank you for pointing this out. We have incorporated additional references across several sections to strengthen the discussions. In particular, we added citations on $\textbf{page 9}$ (revised) to support the discussion of lexical and positional biases.
>
> > Weakness 2: Section 3 describes the sources that contribute to reduced robustness in LLMs. Thus, the appropriate title should be something like “Sources of Robustness Failures in LLMs”.
>
> Thank you for the suggestion. We agree that the revised title more accurately reflects the content of $\textbf{Section 3}$. We have updated the section title to $\textit{“Sources of Non-Robustness.”}$
>
> > Weakness 3: The paper discusses “Accuracy Drop” in section 5.1.1 and the “IID-OOD Performance Gap” in section 5.1.2, but their definitions seem very similar. Please clarify the distinction between these two metrics, as they currently appear redundant.
>
> Thank you for raising this point. We have refined the definition of accuracy drop in $\textbf{Section 5.1.1}$ to better distinguish it from the IID–OOD performance gap discussed in $\textbf{Section 5.1.2}$, thereby clarifying the distinction between the two metrics, with the changes highlighted in violet.
>
> > Typo 1: I understand the motivation behind including Figure 2 and Figure 3; however, on their own, these figures do not clearly convey the intended points. One still needs to refer to the text description to understand their purpose. ...
>
> Thank you for the constructive feedback. The original tree-trunk figure $\textbf{(Figure 3)}$ has been removed due to redundancy with the content of $\textbf{Figure 4}$. The revised $\textbf{Figure 3}$ (previously Figure 4) now includes a descriptive title, $\textit{“Enhancing LLM Robustness: Synergies and Trade-offs,”}$ to clearly communicate the underlying relationships and trade-offs in building robust LLMs. This figure has been streamlined to be self-explanatory and convey the interdependent nature of the robustness dimensions discussed.
>
> > Typo 2 - Future Work says "Will be updated later", which appears to be a placeholder. Please update this appropriately.
>
> Thank you for pointing this out. We have updated $\textbf{Figure 1}$ to replace the placeholder and reflect the finalised content for the $\textbf{Future Work}$ section in the taxonomy.
>
> > Typo 3. The heading Inference Related does not have any accompanying description or content.
>
> Thank you for pointing this out. We have added a description under the $\textbf{Inference-related}$ heading on $\textbf{Page 9}$, clarifying that it covers vulnerabilities arising during model deployment, including issues from decoding strategies, dependence on external retrieval systems, and exposure to distribution shifts or adversarial inputs at inference time.
>
> > Typo 4: page 9 – Phrasing issue: “The process of training and aligning LLMs can introduce such vulnerabilities:” should be revised.
>
> Thank you for pointing out the phrasing issue in $\textbf{page 10}$. We have revised the sentence for clarity and correctness. The updated text now reads: “The process of training and aligning LLMs, therefore, introduces several potential vulnerabilities, including:”
>
> > Typo 5: Section 3.2.2 – 2nd line – the work robustness sources seem weird: “the process itself can introduce robustness Sources in the following ways”. Please rephrase for clarity.
>
> Thank you for highlighting the awkward phrasing. We have revised the sentence for clarity in $\textbf{page 11}$. The updated text now reads: $\textit{“While Reinforcement Learning from Human Feedback (RLHF) is crucial for making LLMs safer and more helpful, the process itself can hinder robustness in the following ways:”}$
> This change eliminates the confusion and conveys the intended meaning more precisely.
>
> > Typo 6: Page 31: Robust Evaluation Strategies -> the following passage has unexplained numbers (I believe the authors wanted to cite these subsections) like 52, 59 etc. in the text. Please fix this.
>
> Thank you for pointing this out. The unexplained numbers were unintended and have now been corrected. We revised the passage at the end of $\textbf{Section 6}$ to properly reference the corresponding subsections instead of displaying stray numbers (e.g., 52, 59). This resolves the issue and improves clarity for the reader.

---

> > ### Author Response · Authors · 2025-09-30
> > **Response to Reviewer LBK7- part2**
> >
> > Other changes are resolved in the above weakness and typo revisions.
> >
> > > Requested changes 1: Please propose a composite benchmark (like a leaderboard) that can be used to evaluate the robustness of open-source LLMs across the dimensions mentioned in the survey. Evaluate at least a few of the popular open-source models (at least the smaller ones) on this new benchmark.
> >
> > Thank you for this thoughtful suggestion. We agree that a composite benchmark would be a valuable contribution to the field. That said, we see this as somewhat beyond the current scope of our survey, which is focused on surveying existing literature rather than conducting new empirical evaluations. Developing such a benchmark would also involve significant design choices like selecting metrics, defining weighting schemes, and setting evaluation protocols, which we believe deserve a dedicated study of their own. Additionally, building and running such benchmarks would require computational resources we don't currently have access to.
> > We had previously highlighted the importance of comprehensive robustness benchmarks and suggested this as a promising direction for future work (Section 6.2, point 6, page 35).

---

### Review · Reviewer_86sW · 2025-09-20

**Summary Of Contributions:**

The authors present an elaborate survey of robustness in LLMs, covering:
- the definition of robustness
- the sources of lack of robustness
- elaborate taxonomy of mitigation strategies
- evaluation metrics and benchmarks-
- discussions on pending research directions.

**Audience:**

Yes

**Broader Impact Concerns:**

Since this is a survey paper, I don't have particular ethical concerns. The authors could include an impact statement in the paper if this is part of TMLRs requirement for publishing.

**Claims And Evidence:**

Yes

**Requested Changes:**

Please "Scope for improvement" above.

## Other stylistic comments
- *Section 2.2*: I am not a big fan of the **label-description** format. This would work well if you have a single phrase following the  “Appearance” and “Cause” labels. However, in some cases, you have multiple descriptive phrases following each heading, making the section look like informally-written study notes. Furthermore, I am not sure if the label “Appearance” best describes how the particular lack of robustness manifests in applications; perhaps “symptoms” might be a better choice of heading?
- *Section 4.2, Adversarial Training*: You may want to clarify here and elsewhere if the losses discussed apply to the **pre-training, SFT or RL stages**
- *Section 4.2, Regularization*: It wasn’t clear to me why contrastive losses and focal losses are clubbed together with regularization approaches. Is there a different categorization you could give to these methods, perhaps “Regularization and loss modifications” (although the previous adversarial training may also fit into to this then)

## Minor comments and typos
- Figure 2 occupies a lot of space, but adds little additional information/insights. Can you make it smaller and more concise?
- “LLMs have been observed to consistently outperform on U.S. datasets compared to Chilean datasets” -> Please provide reference for this
- Figure 3: “This visualization demonstrates how these factors interconnect” → where in the figure are the interconnections highlighted? It looks like the same textual description in the comment is repeated in the figure accompanied by icons.
- Section 3.1.2: “ For instance, crowd-sourced datasets, used widely for fine-tuning and evaluation, frequently contain superficial annotation patterns, where human annotators unintentionally introduce simplistic labeling strategies that models exploit as shortcuts.” → Isn’t this an example of a spurious correlation, or by simplistic labeling strategies, are you referring to strategies that superficially use a protected group for labeling? Please also provide a reference for this.
- Section 3.2.1: I think the critique in Nagarajan et al., 2024 is more about the use of the *next-token prediction* loss, and less to do with optimizing the “average” loss.
- Section 3.2.4, Fine-tuning Limitations: You may want to clarify that these are potential limitations of *vanilla fine-tuning*, not fine-tuning in general. Indeed there are robust versions of fine-tuning that seek to address some of these issues.
- Section 4.1, Page 13, Data Filtering: “LLMS” → “LLMs”
- Section 4.2, Page 14, “uses an entropy regularize” → regularizer?
- Section 4.2, Page 14, “ Xhonneux et al. (2024) proposes” -> propose?
- Section 4.2, Page 14, You mention “DPO” with no context. May want to provide the expansion: “Direct Preference Optimization”, and mention that you are referring to RL methods here
- Section 4.2, Regularization: When mentioning “Debiased Focal Loss”, you may want to clarify if these losses are for typical classification tasks or for general language model generation tasks (my reading of the reference is that it is for the former).
- Section 4.2, Regularization: Many of the fairness references mentioned don’t directly address language modeling. Are you implying that these methods can be adapted to LMs?
- Section 4.2, Alignment, Page 16: Please expand “RPO” → is it Reward-aware Preference Optimization?
- Section 4.3, “This approach provides a cost-effective way … “ → consider splitting it into two sentences (especially after the “though” part).
- Section 4.3, Weight re-distribution, “Zhong et al. (2025b) proposes a framework which ..” → please consider rephrasing this long and overly complicated sentence. I don’t understand what a “pipeline bubble” is.

*Additional reference*: The following is an award-winning work on safety alignment that may be worth including:  Qi et al., Safety Alignment Should Be Made More Than Just a Few Tokens Deep, ICLR 2025.

**Strengths And Weaknesses:**

**P.S.** I would urge the authors to please proof-read the paper. Currently, the paper has *numerous typos* and in some places, *incomplete sentences*. For example, in Figure 1, there is an incomplete block with the phrase “Will be updated later”. Similarly, in Section 3, there is an empty bulleted heading “Inference-Related:” followed by a blank.

___
## What I liked about the paper
- It is timely and has a well-defined scope
- It presents a *clear taxonomy* of the work in this extensive literature
- It provides a nice overview of metrics and benchmarks used to diagnose failure and evaluate robustness, which is bound to be helpful to practitioners
- It has useful discussions on the pending problems in the field
___
___

## Scope for improvement
I am not expert on robustness, and will largely focus on the writing / stylistic aspects. Unfortunately, I find that the writing and the structure of the paper has a lot of scope of improvements, some of which I highlight below.

- **Lack of concrete examples**: The survey would have been more compelling had it provided textual examples (with prompts and responses) to illustrate instances where LLMs lack robustness. For instance, you could provide explicit examples explaining how LLMs can be sensitive to minor prompt variations, or are susceptible to jailbreaking (Sec 2.2). You could explain spurious correlations by illustrating through an example of an LLM learning a superficial pattern instead of the intended complex relationship (Sec 3.1.1).
- **The “Sources” and “Mitigation Strategies” sections seem disconnected:** The paper does a good job of individually categorizing different sources of non-robustness, and categorizing mitigating strategies. However, these two sections seem like separately-written surveys, with the details of the which method caters to which robustness type being hidden in the text. While a clean mapping may not always exist, to the extent possible, the authors could make it clear which mitigation strategies are best suited for a particular type of robustness. In some cases, this is obvious: e.g., sensitivity to prompt variations may be tackled through adversarial training; in other cases, like hallucinations, the solutions are not clearly laid out (e.g. use of RAG or judge LLMs for verification?)
- **Redundancies:** The paper does have occasional redundancies that could be removed.  For example, Section 2.1 lists out common themes in the robustness literature, and Section 2.2 again lists a similar categorization of desired capabilities, with slightly different phrasing and more details. I would find this text cleaner if Section 2.1 could be shortened (without an elaborate list) and have the details deferred to Section 2.2. Similarly, the paper has three separate sections for “Discussion”, “Challenges and Future Work” and “Conclusions”. Combining some of these into smaller subsections may make for a more concise presentation.
- **Math expressions:** The paper completely shies away from using any math expression. While I understand why the authors may want to keep the descriptions simple, in some places, math equations go a long way in stating an idea clearly. For example, when describing Adversarial Training in Section 4.2, it would be easier for the reader to have the loss expressed as an equation, rather than have the inner and outer loops of the adversarial mechanism described in words. Please don’t hesitate to use math expressions wherever they are useful to convey a high-level idea.

---

> ### Author Response · Authors · 2025-09-30
> **Response to Reviewer 86sW**
>
> Comment:
>
> We sincerely thank you for your thoughtful review and constructive feedback. We greatly appreciate your positive comments on the paper’s structure and readability, and we have made revisions to address the identified weaknesses, with the changes highlighted in violet. We also appreciate your recommendation of Qi et al. (2025), which we have incorporated into $\textbf{Section 4}$. Below, we have provided changes corresponding to each of your suggestions as follows:
>
> > Improvement 1: Lack of concrete examples: ...
>
> Thank you for the suggestion. We have added $\textbf{Figure 4  in Section 3}$, which presents concrete textual examples of robustness failures, including sensitivity to prompt variations, jailbreak susceptibility, and spurious correlations. These examples complement $\textbf{Sections 2.2 and 3.1.1}$ by illustrating how LLMs may exploit superficial patterns rather than capturing the intended relationships.
>
> > Improvement 2: The “Sources” and “Mitigation Strategies” sections seem disconnected ...
>
> Thank you for pointing this out. We agree that the connection between the $\textbf{“Sources”}$ and $\textbf{“Mitigation Strategies”}$ sections could be made more explicit. As you have pointed out, to address this, we added a summary table at the $\textbf{end of Section 4}$ that maps specific robustness challenges (e.g., prompt sensitivity, hallucinations, annotation bias) to corresponding mitigation strategies (e.g., adversarial training, retrieval-augmented generation, judge LLMs, data filtering). The table highlights the most effective approaches identified in the literature for each robustness issue. This addition strengthens the linkage between $\textbf{Sections 3 and 4}$ and makes the mitigation landscape clearer and more actionable.
>
> > Improvement 3: Redundancies: The paper does have occasional redundancies that could be removed. ...
>
>  Thank you for the suggestion. We chose to retain $\textbf{Section 2.1}$ in its current form because it serves a distinct purpose: it synthesizes how different authors in the literature conceptualize robustness and, based on these perspectives, motivates the working definition we adopt in the paper. We believe shortening this section would dilute that framing. $\textbf{Section 2.2}$ then builds on this foundation by discussing the dimensions of robustness in greater detail, presented in a narrative style rather than the label-description format to reduce redundancy.
> Regarding the later sections, we streamlined the structure by incorporating some of the discussion points into the introduction and consolidating the $\textbf{“Challenges and Future Work”}$ section for conciseness. The revised manuscript highlights these changes in violet for transparency.
>
> > Improvement 4: Math expressions: The paper completely shies away from using any math expressions. ...
>
> Thank you for the suggestion. In response, we have incorporated mathematical expressions for several of the loss functions, including in $\textbf{Section 4.2 on Adversarial Training and Regularization}$, to improve clarity and precision. However, adding explicit loss formulations for every method discussed would require introducing a number of additional terms and definitions, which risks overwhelming the reader and compromising conciseness. To balance rigour with readability, we included equations where they most directly aid understanding and relied on concise textual descriptions elsewhere to convey the core ideas. We also updated both sections with the changes highlighted in violet.
>
> > Requested changes 1: I am not a big fan of the label-description format. This would work well if you have a single phrase following the “Appearance” and “Cause” labels. ...
>
> Thank you for your suggestion. We have revised the section from a label–description format to a paragraph style to improve readability and ensure a more formal presentation, with the changes highlighted in violet.
>
> > Requested changes 2: Section 4.2, Adversarial Training: You may want to clarify here and elsewhere if the losses discussed apply to the pre-training, SFT or RL stages
>
> Thank you for the comment. We have clarified in $\textbf{Section 4.2}$ that the adversarial training (AT) losses discussed can be applied at different stages depending on the specific method. For example, ALUM (Liu et al., 2020) applies to both pre-training and fine-tuning stages, while methods such as C-AdvUL and CAPO are primarily designed for fine-tuning or alignment stages (SFT/RLHF variants). Where relevant, we have indicated the stage at which each loss is applied to improve clarity for the reader.

---

> > ### Author Response · Authors · 2025-09-30
> > **Response to Reviewer 86sW - part 2**
> >
> > > Requested changes 3: Section 4.2, Regularization: It wasn’t clear to me why contrastive losses and focal losses are clubbed together with regularization approaches. Is there a different categorization you could give to these methods, perhaps “Regularization and loss modifications” (although the previous adversarial training may also fit into this, then)
> >
> > Thank you for the comment. We agree that contrastive and focal losses go beyond classical notions of regularization. To address this, we have revised the section title to $\textbf{“Regularization and Constraint Optimization”}$. We kept this as a separate categorization from adversarial training because these methods operate by reshaping the optimization landscape, through loss modifications or explicit constraints, rather than by augmenting the training data with adversarial examples. This distinction allows us to clearly differentiate approaches that regularize or constrain the learning objective itself from those that inject adversarial perturbations during training.
> >
> > > Requested changes 4: Figure 2 occupies a lot of space, but adds little additional information/insights. Can you make it smaller and more concise?
> >
> > Thank you for the comment. $\textbf{Figure 2}$ has been removed due to redundancy with $\textbf{Figure 3}$. This change helps streamline the presentation while retaining all essential information in the revised figures.
> >
> > > Requested changes 5: “LLMs have been observed to consistently outperform on U.S. datasets compared to Chilean datasets” -> Please provide a reference for this
> >
> > Thank you for the comment. We have added references in $\textbf{Section 2.2.5}$ to support the observation that LLMs often perform better on U.S.-based datasets compared to Chilean datasets, highlighting geographical bias and the differing influence of socio-demographic attributes on model performance (Abeliuk et al., 2025; Qu & Wang, 2024).
> >
> > > Requested changes 6: Figure 3: “This visualization demonstrates how these factors interconnect” → where in the figure are the interconnections highlighted? It looks like the same textual description in the comment is repeated in the figure, accompanied by icons.
> >
> > Thank you for the comment. We have reduced the caption of $\textbf{Figure 3}$ and moved the discussion of the interconnections between factors to the concluding text of $\textbf{section 2 on Page 8}$, making the figure itself more concise and allowing the relationships to be explained clearly in context.
> >
> > > Requested changes 7: Section 3.1.2: “ For instance, crowd-sourced datasets, ... .Please also provide a reference for this.
> >
> > Thank you for the comment. We have clarified that the example regarding crowd-sourced datasets illustrates annotation bias, which can create spurious correlations by causing models to rely on biased labelling patterns. However, we note that not all spurious correlations arise from annotation bias, as some of them stem from naturally occurring patterns in the data that do not reflect the task. Accordingly, we retained this discussion in the $\textbf{“Dataset Biases and Anomalies”}$ section, and provided references to support the impact of annotation bias and benchmark design on robustness (Thrampoulidis, 2024; Nagarajan et al., 2024).
> >
> > > Requested changes 8: Section 3.2.1: I think the critique in Nagarajan et al., 2024 is more about the use of the next-token prediction loss, and less to do with optimizing the   “average” loss.
> >
> > Thank you for the comment. We have revised $\textbf{Section 3.2.1}$ to clarify that the critique primarily concerns the use of the next-token prediction (NTP) loss rather than the optimization of the average loss. The updated text highlights how NTP can encourage reliance on superficial patterns in the training data, potentially limiting generalization and robustness, while still enabling the model to capture syntactic, semantic, and contextual dependencies (Thrampoulidis, 2024). All changes are highlighted in violet, with the original text shown in strikethrough.
> >  > Requested changes 9: Section 3.2.4, Fine-tuning Limitations: You may want to clarify that these are potential limitations of vanilla fine-tuning, not fine-tuning in general. Indeed, there are robust versions of fine-tuning that seek to address some of these issues.
> >
> > Thank you for the suggestion. We have clarified in $\textbf{Section 3.2.4}$ that the discussed limitations pertain to vanilla fine-tuning and do not apply to fine-tuning in general. We also briefly note that there exist robust fine-tuning approaches designed to address some of these issues in $\textbf{Section 4}$.
> >
> > > Requested changes 10: Section 4.1, Page 13, Data Filtering: “LLMS” → “LLMs”
> >
> > Thank you for the comment. We have corrected the typo in $\textbf{Section 4.1 (Page 15)}$, changing “LLMS” to “LLMs.”

---

> > > ### Author Response · Authors · 2025-09-30
> > > **Response to Reviewer 86sW - part3**
> > >
> > > > Requested changes 11: Section 4.2, Page 14, “uses an entropy regularize” → regularizer?
> > >
> > > Thank you for noticing this. Upon careful review, we found that the sentence was incorrect and have removed it entirely from $\textbf{Section 4.2, Page 17}$.
> > >
> > > > Requested changes 12: Section 4.2, Page 14, “ Xhonneux et al. (2024) proposes” -> propose?
> > >
> > > Thank you for pointing these out. We have corrected “uses an entropy regularize” to “uses an entropy regularizer” and updated “Xhonneux et al. (2024) proposes” to “Xhonneux et al. (2024) propose” in $\textbf{Section 4.2, Page 17}$, as well as in several other instances throughout the manuscript.
> > >
> > > > Requested changes 13: Section 4.2, Page 14, You mention “DPO” with no context. May want to provide the expansion: “Direct Preference Optimization”, and mention that you are referring to RL methods here
> > >
> > > Thank you for your comment. We have updated Section 4.2 (page 19) to expand “DPO” as Direct Preference Optimization and clarified that it refers to RL-based methods. Additionally, we provide context on its purpose and relationship to other alignment techniques, ensuring readers understand its role in improving model robustness, with the changes highlighted in violet.
> > >
> > > > Requested changes 14: Section 4.2, Regularization: When mentioning “Debiased Focal Loss”, you may want to clarify if these losses are for typical classification tasks or for general language model generation tasks (my reading of the reference is that it is for the former).
> > >
> > > > Requested changes 15: Section 4.2, Regularization: Many of the fairness references mentioned don’t directly address language modelling. Are you implying that these methods can be adapted to LMs?
> > >
> > > Thank you for your comment. We have completely revised $\textbf{Section 4.2 on Regularization}$ to include more recent methods and, where applicable, provide the corresponding mathematical formulations. The updated section also highlights approaches such as Debiased Focal Loss (DBL) (Karimi Mahabadi et al., 2020), confidence regularization (Utama et al., 2020), and lexical co-occurrence-based regularizers (Garimella et al., 2021). While many of these methods were initially developed for classification tasks rather than generative language models, their underlying principles, such as adaptive reweighting to reduce reliance on spurious correlations, penalizing sensitivity to biased features, and adjusting loss functions to handle noisy data, are conceptually transferable to LLMs. Collectively, these methods demonstrate how structured regularization can be leveraged to balance robustness, fairness, efficiency, and generalization in modern language models.
> > >
> > > > Requested changes 16: Section 4.2, Alignment, Page 16: Please expand “RPO” → is it Reward-aware Preference Optimization?
> > >
> > > Thank you for the comment. We have expanded $\textbf{Section 4.2 on Alignment at pages 18-19}$ to clarify all abbreviations, including “RPO,” which now appears as “Reward-aware Preference Optimization”. Additionally, we have provided a more detailed explanation throughout the section for improved clarity.
> > >
> > > > Requested changes 17: Section 4.3, “This approach provides a cost-effective way … “ → consider splitting it into two sentences (especially after the “though” part).
> > >
> > > Thank you for the suggestion. We have revised the sentence in $\textbf{Section 4.3}$ by splitting it into two sentences, particularly around the “though” segment, to improve readability and clarity, with the changes highlighted in violet.
> > >
> > > > Requested changes 18: Section 4.3, Weight re-distribution, “Zhong et al. (2025b) proposes a framework which ..” → Please consider rephrasing this long and overly complicated sentence. I don’t understand what a “pipeline bubble” is.
> > >
> > > Thank you for your feedback. We have rephrased the sentence in $\textbf{Section 4.3 on Weight Redistribution}$ for clarity and conciseness. Additionally, we have clarified the concept previously referred to as a “pipeline bubble” to ensure its meaning is understandable to the reader, with the changes highlighted in violet.

---

> > > > ### Comment · Reviewer_86sW · 2025-10-28
> > > > **Thanks a lot for addressing my comments**
> > > >
> > > > I greatly appreciate the authors for the detailed response and addressing each of my concerns.
> > > >
> > > > I have a few follow-up comments.
> > > >
> > > > **Concrete examples:** Thanks for including examples of lack of robustness in Figure 4. If possible, could you also mention in the caption, the specific LLM family that was used to elicit these responses?
> > > >
> > > > **Connection between Section 3 and 4:** Thanks for including a table (Table 3) mapping challenges to mitigation strategies. Could you add a line in the main text pointing to this table?
> > > >
> > > > Also, it would be good to add some text at the beginning of Section 4, so that the transition from Section 3 to 4 is smooth. For example, currently, Section 4 begins with the sentence “The strategies are categorized based on the stage of …”. Instead could you start with something like “Having covered the various challenges in robustness, we will now look into mitigation strategies ….”
> > > >
> > > >
> > > > **Other changes:** Thanks for the restructuring of the conclusion section and incorporation of math notation wherever relevant/useful.
> > > >
> > > > **Typo:** Missing space in Page 9:  “training and deployment.Figure 4”.
> > > >
> > > > *Minor request*: can you increase spacing between the rows in Table 3, so that the different threat-mitigation strategies blocks can be distinguished from one another?
> > > >
> > > >
> > > > You may also want to change “Constraint Optimization” to “Constrained Optimization”.

---

> > > > > ### Author Response · Authors · 2025-10-29
> > > > > **Response to Reviewer 86sW comments**
> > > > >
> > > > > Thank you for your comments. We have revised the paper as advised and updated the PDF as well.
> > > > >
> > > > > > $\textbf{Figure 3 Caption}$
> > > > >
> > > > > Thank you for the suggestion. We've updated the caption accordingly.
> > > > > > $\textbf{Table 3 reference:}$
> > > > >
> > > > > Added a line in the 2nd paragraph of Section 4 in page 14.
> > > > > >$\textbf{Connection between Section 3 and 4: }$
> > > > >
> > > > > Added a line in the intro of Section 4 in page 14
> > > > > >$\textbf{Missing space in page 9:}$
> > > > >
> > > > > Issue Resolved
> > > > > >$\textbf{Table 3 Spacing:}$
> > > > >
> > > > > Added a line between each row in table 3 in page 22
> > > > > >$\textbf{Typo:}$
> > > > >
> > > > > "Constraint Optimization" is changed to "Constrained Optimization"

---

### Author Response · Authors · 2025-09-30
**Revision**

We thank all the reviewers for their comments and insights. We have now uploaded a revised version of our manuscript where we addressed your comments. Specifically, we have:

- Fixed some typos and made minor improvements
- Added survey methodology
- Refined Section 4.2
- Added a table for the cost-benefit analysis of different mitigation strategies
- Added a flowchart for actionable guidance to LLM practitioners
- Removed the discussion section and added important information in the intro

---

### Public Comment · ~Anwoy_Chatterjee1 · 2025-11-11
**Metrics quantifying prompt sensitivity**

Dear Authors, Reviewers and Action Editor,

It's great to see such a detailed survey on the robustness of LLMs. I would also like to congratulate and thank the authors for the great work. As I read the paper, I noticed that the authors mentioned "**Sensitivity to Prompt Variations**" as a key dimension of robustness (Section 2.2.3). But I find the discussion on this robustness issue to be very limited in the paper. This is probably because the keywords authors use to gather the works somehow missed a significant body of work in this field. For instance, many works demonstrate and analyse the issue of "prompt sensitivity" in LLMs [1, 2] and even define a metric to quantify it [3, 4]. *POSIX* [3], for example, is a performance-agnostic metric that quantifies the robustness of LLMs to "intent-preserving" prompt variations - which could have been discussed under Section 5.1.3 (Consistency Metrics) of the paper. **I believe the inclusion of some of the major works from this significant body of works on prompt sensitivity of LLMs will enhance the quality and richness of the survey**.

Congrats again on such a nice piece of work.

[1] Melanie Sclar, Yejin Choi, Yulia Tsvetkov, and Alane Suhr. 2023. Quantifying language models’ sensitivity to spurious features in prompt design or: How i learned to start worrying about prompt formatting. In ICLR 2024.

[2] Moran Mizrahi, Guy Kaplan, Dan Malkin, Rotem Dror, Dafna Shahaf, and Gabriel Stanovsky. 2024. State of What Art? A Call for Multi-Prompt LLM Evaluation. Transactions of the Association for Computational Linguistics, 12:933–949.

[3] Anwoy Chatterjee, H S V N S Kowndinya Renduchintala, Sumit Bhatia, and Tanmoy Chakraborty. 2024. POSIX: A Prompt Sensitivity Index For Large Language Models. In Findings of the Association for Computational Linguistics: EMNLP 2024, pages 14550–14565, Miami, Florida, USA. Association for Computational Linguistics.

[4] Jingming Zhuo, Songyang Zhang, Xinyu Fang, Haodong Duan, Dahua Lin, and Kai Chen. 2024. ProSA: Assessing and Understanding the Prompt Sensitivity of LLMs. In Findings of the Association for Computational Linguistics: EMNLP 2024, pages 1950–1976, Miami, Florida, USA. Association for Computational Linguistics.

---

### Decision · Action_Editor_iP8Q · 2025-11-02

**Recommendation:** Accept as is

**Additional Comments:**

Overall, the survey is a valuable contribution to the literature. One critique from reviewers that we tend to agree with is that the brevity could be improved, perhaps with greater reliance on mathematical rather than textual descriptions. The authors have somewhat remedied this for the adversarial training section, but ideally this could permeate to even the broad overview of the problem space. We acknowledge that this may partly be a matter of taste, though we would encourage the authors to consider if even the textual discussion could be compressed so as to make it easier for interested readers to glean the main findings.

**Audience:**

Yes

**Audience Explanation:**

As made clear in the paper, there is a large body of recent work studying various aspects of LLM robustness; the area is thus of clear interest to the community. The paper's contribution of providing a systematic overview of this literature, and identifying some gaps for future works, can reasonably be expected to be of interest to the community. This was also the opinion of all reviewers.

**Claims And Evidence:**

Yes

**Claims Explanation:**

The paper's primary contribution is providing a survey of recent works on LLM robustness, including a categorisation of sources of non-robustness and mitigation strategies for the same. These are synthesised to highlight some common trends, and identify areas for future work.

Reviewers were unanimous in appreciating the paper's systematic treatment of the area. Two lacunae that were identified were the occasional lack of specific examples, and a lack of clarity on the precise process for identifying relevant work. These were remedied to the reviewers' satisfaction. From our reading, we concur with the reviewers' positive view on this point.